# The P protein T25M substitution is involved in the quasispecies and virulence of Newcastle disease virus

Prince-Théodore Daguia-Wenam,[1,2,3] Kejia Lu,[1] Xueting Zhou,[1] Chuanqi Yan,[4] Lina Tong,[5] Haijin Liu,[1] Zengqi Yang[1]

**ABSTRACT** The non-virulent Newcastle disease virus (NDV)-Spotted Dove strain acquires enhanced virulence after serial passage in chicken embryos. In this study, the P gene T25M substitution was identified as the critical determinant of increased pathogenicity after 100 consecutive passages in specific-pathogen-free (SPF) chicken embryos. Initial characterization of the parental Spotted Dove strain confirmed a lentogenic phenotype by mean death time (MDT) in 9-day-old SPF embryos, intracerebral pathogenicity index (ICPI) in 1-day-old chicks, intravenous pathogenicity index (IVPI) in 6-week-old chickens, limited replication in DF-1 cells, and minimal tissue pathology. The virus exhibited shortened MDT, elevated ICPI and IVPI scores, higher replication in DF-1 and BHK-21 cells, and extensive lesions in respiratory and neural tissues. Next-generation sequencing of five viral RNA samples from the parental strain and selected passage points revealed six nonsynonymous mutations, with only T25M in P correlating with the virulence phenotype. To validate its role, recombinant NDV-Dove strains (rDove) and rDove$^{100th}$-P$_{T25M}$ were generated and co-inoculated into 9-day-old SPF chicken embryos at $1 \times 10^4$ PFU/mL. Similarly, rDove and rDove$^{100th}$-P$_{T25M}$ containing green and red fluorescence, respectively, were co-infected into DF-1 cells at a multiplicity of infection of 0.01 with equal ratios (1:1, 1:9, and 9:1). After 10 passages, results indicated that multiple factors equalized the quasispecies frequencies as the initial differences between "C" and "T" were gradually overcome by the evolutionary processes of the viral population in chicken embryos and DF-1 cells. Our findings identify P-T25M as the key adaptive mutation driving enhanced NDV virulence and illustrate the complex quasispecies evolution during serial passage.

**IMPORTANCE** This research investigated the T25M mutation in the P gene of the Newcastle disease virus (NDV)-Spotted Dove strain, which emerged as a significant mutation conferring virulence after 100 serial passages through chicken embryos. The P gene typically encodes a viral phosphoprotein that serves various roles in the viral life cycle, including involvement in viral RNA synthesis, interaction with other viral proteins, and occasionally modulation of the host immune response. A T25M mutation, characterized by the substitution of threonine with methionine at position 25, has the potential to modify the structural configuration of the P gene. The findings highlight the significance of the extensive quasispecies population of NDV, which develops virulence and improves viral replication and transmission in chickens. Key point mutations, particularly T25M in these cases, are essential for viral fitness and the evolution of NDV quasispecies in chicken embryos and DF-1 cells.

**KEYWORDS** Newcastle disease virus, chicken embryos and chickens, pathogenicity, virulence, next-generation sequencing

Newcastle disease virus (NDV), classified under the Avulavirus genus of the Paramyxoviridae family, demonstrates a spectrum of virulence (1), from

**Peer Reviewers** Wentao Li, Huazhong Agricultural University, Wuhan, Hubei, China; Ravindra P. Veeranna, Xavier University School of Medicine, Oranjestad, Aruba

Address correspondence to Haijin Liu, liuhaijin@nwafu.edn.cn, or Zengqi Yang, yzq8162@126.com.

The authors declare no conflict of interest.

See the funding table on p. 24.

asymptomatic infections to highly pathogenic variants that lead to significant disease in poultry and other avian species. The virulence of the virus is determined by a combination of viral genetic factors and host immune responses, with several key genes being essential in both viral pathogenesis and the evasion of host defenses (2). Hemagglutinin neuraminidase (HN) and fusion (F) proteins are critical virulence factors that significantly contribute to immunogenicity (3). NDV virulence is a multigenic characteristic, with viral genes such as F, HN, and P acting as key determinants of pathogenicity, while host immune response genes influence susceptibility and disease severity. The NDV genome typically consists of a 55-nucleotide leader at the 3′ end and a 114 nucleotide trailer at the 5′ end, surrounding six essential genes that encode the nucleocapsid (N), matrix protein (M), phosphoprotein (P), fusion protein (F), HN protein, and large polymerase protein (L), with particular genes associated with virulence (4). The F gene encodes the fusion protein that facilitates viral entry into host cells by promoting the fusion of the viral envelope with the host cell membrane. The proteolytic cleavage site of the F protein serves as a key factor influencing virulence. Strains exhibiting high virulence (velogenic) include a multibasic cleavage site ($^{112}$(R/K)-R-(Q/K/R)-(R/K)-R-F$^{117}$) (5), which is acknowledged by widespread cellular proteases, facilitating systemic infection. Intermediate (mesogenic), low pathogenic (lentogenic), or avirulent strains are characterized by their low pathogenicity and reduced virulence, possess a monobasic cleavage site ($^{112}$(G/E)-(R/K)-Q-(G/E)-R-L$^{117}$) (6). This cleavage occurs exclusively through specific proteases (e.g., in the respiratory or intestinal tract), thereby restricting tissue tropism. The HN protein facilitates viral attachment to host cell receptors through hemagglutinin activity and promotes the release of progeny virions via neuraminidase activity. Mutations in HN can induce alterations. Receptor binding specificity affects cell tropism and tissue invasion. The efficiency of neuraminidase influences viral dissemination and immune evasion through the disruption of host mucin barriers. The P protein serves multiple functions as an accessory protein, playing roles in viral replication, transcription, and immune evasion. Essential functions comprise functions as a cofactor for the viral polymerase complex (P-N-L) to enable genome replication. Inhibition of transcription factors, such as IRF3 and STAT proteins, disrupts host interferon (IFN) signaling. Mutations in the P gene can enhance virulence by altering protein structure, influencing interactions with host or viral factors, or suppressing IFN, as recently reviewed (7). The L protein functions as the viral RNA-dependent RNA polymerase. Mutations in L, while less characterized for virulence compared to F or P, can impact replication efficiency and transcription fidelity, thereby indirectly affecting viral fitness and pathogenicity (8, 9). Phylogenetic analyses indicate that the majority of NDV strains obtained from pigeons and doves are classified within genotype VI (4, 10–12).

Virus evolution has been examined primarily in the context of disease emergence, lacking integration into the wider field of evolutionary biology (13, 14). While virus evolution shares similarities with host evolution, it is not identical. In RNA viruses, high mutation rates give rise to the quasispecies concept, in which viral populations composed of a diverse range of variants serve as the basis for evolutionary processes (15). This diversity enhances a virus's capacity for adaptation. However, some RNA viruses exhibit remarkable evolutionary stability. In large population passages and depending on the biological within a constant environment, fitness typically increases (16, 17). The quasispecies concept applies to various biological entities, particularly in instances of small genome size and enhanced mutation rates, as observed in RNA viruses, many of which are notable pathogens. Viral infections commence with a population of viral particles instead of an individual virus particle (18). The variants that persist and can reinfect a new host indicate selective pressures acting externally to the host. Furthermore, variation is produced through recombination and reassortment. Viral quasispecies can be represented as clouds in a theoretical sequence space, influenced by replication fidelity and the prevalence of the master sequence. Viral quasispecies denote a collection of genetically related yet distinct genomes that undergo ongoing genetic variation, competition, and selection, functioning as a cohesive unit of selection. This introduces

further layers of competition, including among viral genomes. Viruses from distinct replicative units compete for tissue and organ invasion. The intracerebral pathogenicity index (ICPI) is the widely recognized approach for evaluating the virulence of NDV strains, because of its proven accuracy and sensitivity. The differences in virulence among various NDV isolates are indicated by the ICPI, which spans from 0.0 for non-virulent viruses to 2.0 for highly virulent viruses.

This study aimed to evaluate the viral fitness and quasispecies dynamics of the NDV-Spotted Dove strain. The strain under investigation was isolated from wild birds at the northern foot of the Qinling Mountains in Shaanxi Province, China, in 2008. This strain, which originated from the Spotted-necked Dove, belongs to genotype IX. The strain exhibited an ICPI of 0.425 and a mean death time (MDT) of 64.8 h (19) in Spotted-necked Doves, classifying it as lentogenic. To assess whether this strain remains stable in terms of both virulence and genome after serial passages through chicken embryos, to determine whether the immune response to the strain is maintained after multiple passages, and to evaluate whether both the original and passaged strains can protect chickens from infection by virulent NDV strains. Additionally, the study aimed to examine how these factors contribute to the viral fitness and quasispecies evolution.

## MATERIALS AND METHODS

### Virus and cell

A lentogenic strain of Newcastle Disease virus (NDV), named Spotted-necked Dove, was isolated at the northern foot of the Qinling Mountains in Shaanxi Province, China, in 2008 (19). Nine-day-old specific pathogen-free (SPF) chicken embryos were used for virus purification and were subsequently stored at −80°C in the Veterinary Infectious Disease Laboratory, Northwest A&F University, Shaanxi, China. Chicken embryo fibroblasts (DF-1 cells) and Baby Hamster Kidney (BHK-21) cells were purchased from American Type Culture Collection (ATCC) and were maintained in our laboratory. These cells were cultured in Dulbecco's Modified Eagle Medium (DMEM, Gibco, Grand Island, USA) supplemented with 10% fetal bovine serum (FBS, Gibco, Grand Island, USA) and were also maintained in DMEM containing 1% FBS at 37°C in a 5% $CO_2$ atmosphere for future use.

### Embryos and chickens

The SPF chicken embryos (9 days old), as well as 1-day-old, 3- and 4-week-old, and 6-week-old SPF chickens used in this study, were provided by Jinan SAIS Industry, Shandong Province, China.

### Passage of NDV-Dove strain through chicken embryo and plaque purification test

A total of 100 serial passages were performed in this study. SPF embryonated chicken eggs (SPF ECEs) were used to propagate the original NDV-Dove strain (parental) through allantoic fluid, as previously described (20). The virus was inoculated into 9-day-old SPF chicken embryos for propagation and incubated at 37°C to allow viral development within the ECE, with daily monitoring of the eggs. Three to four days post-inoculation (dpi), or when embryo death occurred, the eggs were stored overnight at 4°C. Virus was then collected from the allantoic fluid, and the viral titer was determined using the hemagglutination assay (HA) (21, 22). Positive samples with HA titers of $2^6–2^{8+}$ were pooled, filtered, and stored at −80°C for future use. Virus titer determination through plaque purification was conducted using BHK-21 cells, as previously described (23). For each passage up to the 100th passage, three eggs were used per passage, and each egg was inoculated with 200 µL of a viral suspension containing $1 \times 10^5$ PFU/mL.

## Viral replication test

DF-1 cells were cultured in 24-well culture plates and allowed to grow until they reached 80%–90% confluence overnight. On the day of the experiment, cells were inoculated with two viruses (the original NDV-Dove strain and its 100th passaged strain) at a multiplicity of infection (MOI) of 0.01 in triplicate wells (24). The cells were incubated at 37°C in a 5% $CO_2$ atmosphere for 1-h post-infection (hpi). After 1 h, the cells were washed with phosphate-buffered saline (PBS), and 500 µL of DMEM containing 1% FBS was added to each well. The cells were then incubated at 37°C, 5% $CO_2$ for 4 dpi. Every 24 h, 100 µL of cell supernatant was collected and replaced with an equal volume of DMEM containing 1% FBS. These supernatants were assayed for virus titers using the $TCID_{50}$/mL test in BHK-21 cells.

## Viral pathogenicity index

The purpose of this study was to evaluate and compare the properties of the parental NDV-Dove strain and its 100th passaged strain in *vivo*. Nine-day-old SPF ECEs were used to propagate the virus through the allantoic cavity from the parental strain through to the 100th passage. According to the OIE guidelines (World Organization for Animal Health [WOAH], formerly OIE, 2021 https://www.woah.org), the MDT in 9-day-old SPF chicken embryos, the ICPI in 1-day-old SPF chickens, and the intravenous pathogenicity index (IVPI) in 6-week-old SPF chickens were determined to assess the virulence and replication of both the parental and passaged NDV-Dove strains.

## Animal experiment

The parental NDV-Dove strain and the 100th passaged strain were used to infect 4-week-old SPF chickens to evaluate the pathogenicity, detoxification (25), and replication of the passage of strain after 100 serial passages through chicken embryos. A total of 30 4-week-old SPF chickens were randomly assigned to three groups, including PBS as a negative control. Chickens were selected randomly and were kept in separate isolators, with each group containing 10 chickens. Fresh allantoic fluid diluted in PBS was administered oculonasally with 200 µL per chick containing $5 \times 10^6$ PFU/mL. The first group was infected with the parental NDV-Dove strain, the second group was infected with the NDV-Dove[100th] strain, and the third group, which served as a negative control, respectively, and they were checked twice a day. The general flock condition, including temperature, lighting, water, feed, and litter conditions, was recorded daily. Plastic boots and gloves were discarded in designated containers after use. We were all in a good mood with good weather during the study.

During the observation period, clinical symptoms in chickens, including respiratory signs, depression, and diarrhea, were recorded, noting the onset time, symptom description, and mortality. A survival curve and clinical symptom index were generated. On days 3 and 5 post-inoculation (when chickens exhibited severe clinical symptoms or died), three chickens from each group were sacrificed for necropsy to observe and document pathological changes in various tissues and organs. Blood samples were collected from live chickens on days 7 and 14 dpi for hemagglutination inhibition (HI) titer testing.

## RNA extraction and library preparation

The parental NDV-Spotted Dove strain sequence has been submitted to GenBank (https://www.ncbi.nlm.nih.gov/nuccore/KC934170.1). The parental NDV-Spotted Dove strain, considered as the first passage and named Dove[1st], along with the 25th passage (Dove[25th]), 50th passage (Dove[50th]), 75th passage (Dove[75th]), and 100th passage (Dove[100th]), was selected at predetermined intervals to infect the DF-1 cell to study the syncytia formation. According to microscopic observation, these samples were selected for sequencing and phenotypic assays. The samples were centrifuged at 4°C at $13,000 \times g$ for 3 min. These five viral RNAs were reverse-transcribed into complementary DNA (cDNA)

using Oligo primers to convert the RNA single strands into double strands. The cDNA was then fragmented using PCR with specific primers for 25 cycles, according to the PCR protocol: 98°C for 1 min, 98°C for 20 s, 55°C for 20 s, 72°C for 30 s, followed by 72°C for 5 min and a final step at 16°C for 25 cycles. Sequencing linkers were ligated to both ends of the obtained cDNA fragments. The cDNA library was subsequently amplified by bridge PCR and sequenced on the Illumina NovaSeq 6000 platform using a paired-end 150 bp sequencing strategy. After DNase treatment and cleanup, second-strand synthesis was performed before library preparation using Nextera XT reagents (Illumina). Sequencing was carried out on the NovaSeq 6000 (Illumina). Although initially described as a consensus-level sequencing methodology, the depth of coverage enabled deep sequencing analysis. Bioinformatics analysis was performed using the pipeline previously described.

## Next-generation sequencing process

The processing and data analysis of next-generation sequencing (NGS) for different passaged strains were performed as previously described. Raw reads underwent filtering and trimming via fastp (https://github.com/OpenGene/fastp) to remove sequencing adapters and low-quality reads, including those with a quality score below Q20. Ribosomal RNAs and host reads were subtracted using BBMAP (https://github.com/BioInfoTools/BBMap) by read mapping. *De novo* genome assembly was performed using SPAdes v3.14.1 (https://github.com/ablab/spades). Assembled scaffolds were filtered with a minimum contig length of 100 bases, and the best BLAST hits were identified in the NCBI nt database. The complete genome sequence of the NDV-Spotted Dove strain was obtained and annotated based on available NDV-Spotted Dove strain genomic sequences in the Swiss-Prot database. High-quality filtered reads were classified by Kraken2 (v2.0.8-beta), using precise alignment of kmers of varying lengths.

## Construction of different amino acid mutants affecting key sites during the passage of the NDV-Dove strain and virus rescue

The complete NDV-Dove strain genome was divided into six fragments, each containing a major open reading frame (ORF) flanked by short 5′ and 3′ untranslated regions, followed by conserved transcriptional initiation and termination control sequences known as the gene start and gene end, respectively. These six fragments, including their ORF regions and specific primers, were amplified from viral RNA extracted from the allantoic fluid of NDV-infected ECEs, following the manufacturer's instructions. The viral RNA was reverse-transcribed into cDNA and assembled into the pBR322 vector. This vector was used for the insertion of the virus genome (Leader-NP-P-M-F-HN-L-Trailer) along with appropriate restriction sites. The pBR322 vector and DH10β competent cells are preserved by the Major Disease Prevention and Control and Purification Team at the School of Veterinary Medicine, Northwest A&F University. pDove expression plasmids were generated and amplified in DH10β-competent cells, cultivated in a 30°C incubator with tetracycline DMEM (LB), and purified using the Quik Plasmid extraction kit (OMEGA) following the manufacturer's protocol. The plasmids were extracted and stored at −20°C for future use, and the recombinant virus (rDove) was stored at −80°C.

To investigate the amino acid substitutions in sensitive regions that affect the passage of the NDV-Dove strain through chicken embryos after 100 serial passages, six-point mutations were introduced into infectious cDNA clones. The mutations, including T25M, S369N, S200N, N147D, I405L, and K495N, were generated from $pDove^{100th}$-P, $pDove^{100th}$-M, and $pDove^{100th}$-HN, respectively. Each mutant cDNA was cloned into the pBR322-Dove vector, incorporating specific resistance sites. The resulting plasmids were named $pDove^{100th}$-$P_{T25M}$, $pDove^{100th}$-$P_{S369N}$, $pDove^{100th}$-$M_{S200N}$, $pDove^{100th}$-$HN_{N147D}$, $pDove^{100th}$-$HN_{I405L}$, and $pDove^{100th}$-$HN_{K495N}$, and stored at −20°C for future use. Virus rescue was performed in six-well plates using OriFect Transfection Reagent. Briefly, corresponding cells were co-transfected with 5 µg/well (2 µg plasmid, 2 µg pNPL, 1 µg pT7). After 72 h post-transfection (hpt), infected cell lesions were observed, and the supernatants were

collected. Nine-day-old SPF chicken embryos were used to propagate the recombinant viruses, which were named rDove[100th]-$P_{T25M}$, rDove[100th]-$P_{S369N}$, rDove[100th]-$M_{S200N}$, rDove[100th]-$HN_{N147D}$, rDove[100th]-$HN_{I405L}$, and rDove[100th]-$HN_{K495N}$. The viruses were stored at −80°C after sequencing for future use. The recombinant viruses were grown in DF-1 cells as described previously in viral replication.

## Determining the impact of the key point affecting the passage of NDV-Dove strain through chicken embryos

The recombinant Dove[100th] mutants were inoculated into 3-week-old SPF chickens to evaluate the pathogenicity, detoxification, and replication of the passaged strains after serial passages through chicken embryos. In this study, 80 3-week-old SPF chickens were used. The chickens were randomly assigned to eight groups of 10 chicks each, with one group serving as a negative control and receiving PBS. Fresh allantoic fluid from the six rDove[100th] mutants was diluted in PBS and administered oculonasally at 200 µL containing $5 \times 10^6$ PFU/mL per chick. Groups 1 through 6 were inoculated with the recombinant viruses rDove[100th]-$P_{T25M}$, rDove[100th]-$P_{S369N}$, rDove[100th]-$M_{S200N}$, rDove[100th]-$HN_{N147D}$, rDove[100th]-$HN_{I405L}$, and rDove[100th]-$HN_{K495N}$, respectively, while the last two groups were inoculated with PBS and rDove as control groups. The experimental procedure adhered to animal protocols, as previously described in the section on animal experiments.

## Generated and constructed of the NDV-Dove strain containing the EGFP and mCherry genes

The recombinant NDV-Dove expressing green and red fluorescent proteins (EGFP and mCherry) was generated and amplified in DH10β-competent cells. They were cultivated in a 30°C incubator with tetracycline DMEM (LB), and plasmids were purified using the Quik plasmid extraction kit, according to the manufacturer's protocol. The purified plasmids were stored at −20°C for future use. Briefly, from pDove and pDove[100th]-$P_{T25M}$, the ORF of EGFP and mCherry was inserted between the P-M gene junction using specific primers and restriction sites (PacI and AsiSI), ensuring that the full-length fragment adhered to the six-base insertion principle. The resulting plasmids, pDove-EGFP and pDove[100th]-$P_{T25M}$-mCherry, were obtained. Virus rescue was performed in six-well plates using TurboFect Transfection Reagent. Each well was co-transfected with 5 µg of plasmids (2 µg of pDove, 2 µg of pNPL, and 1 µg of pT7). Fluorescence expression was observed every 24 hpi. At 72 hpi, the cell supernatant was harvested and immediately inoculated into 9-day-old SPF chicken embryos for virus amplification. The HA test was performed to confirm the presence of fluorescence. The recombinant viruses were subsequently named rDove-EGFP and rDove[100th]-$P_{T25M}$-mCherry.

## Co-inoculated a chicken embryo with two different strains

A total of 10 serial passages were performed in this study. SPF chicken embryos were used for the propagation of the original NDV-Dove strain through allantoic fluid. As previously described, the recombinant NDV-Dove strain (rDove) was derived from the parental NDV-Dove strain (Spotted Dove/China/08), and the recombinant NDV-Dove[100th]-$P_{T25M}$ strain (NDV-rDove[100th]-$P_{T25M}$) was obtained from the NDV-Dove[100th] strain, purified, and stored at −80°C in our laboratory. The two viruses were mixed and inoculated into 9-day-old SPF chicken embryos at a concentration of $1 \times 10^4$ PFU/mL using three different dilution ratios (1:1, 1:9, and 9:1) of rDove to rDove[100th]-$P_{T25M}$, respectively. The eggs were incubated at 37°C to allow the virus to develop in the ECE, with checks performed twice daily. Two dpi or when the embryos died. The virus was then collected from the allantoic fluid for virulence titration using a HA. Positive HA titers were mixed, centrifuged at $12,000 \times g$ for 3 min at 4°C, and stored at −80°C for future use. Virus titers were determined by plaque purification in BHK-21 cells, as described in our previous study (23), with each passage continuing until the 10th passage.

## Co-infected the DF-1 cell with a different strain containing green and red proteins

NDV-Dove strains expressing green and red fluorescent proteins were obtained from pDove and pDove100th-$P_{T25M}$ strains, respectively. The viral titers of rDove-EGFP and rDove100th-$P_{T25M}$-mCherry, as well as the reduced titers of rDove and rDove100th-$P_{T25M}$, were determined using the $TCID_{50}$/mL assay. For each passage, 12-well plates were used for cell culture. DF-1 cells were co-infected in an 80% confluent flask with rDove-EGFP and rDove100th-$P_{T25M}$-mCherry at an MOI of 0.01 $TCID_{50}$ per $2\times10^6$ cells, using three different dilution ratios (1:1, 1:9, and 9:1) in three replicates. After 1 h of incubation at 37°C in 5% $CO_2$, the cells were washed twice with PBS and overlaid with infection medium. The viral supernatant was collected at 48 hpi, centrifuged at $12,000 \times g$ for 3 min at 4°C, and then stored at −80°C for subsequent titration. As different glycosylation patterns of viruses processed in cells can influence pathogenesis, this study controlled for cell type by comparing each passage ($P^1$) virus to the $P^{10}$ virus from the same series for viral titers using $TCID_{50}$/mL.

## Statistical analysis

The results of this study were analyzed using GraphPad Prism 9 (GraphPad Software, Inc., San Diego, CA, USA). The data are presented as the mean ± standard deviation (SD), and the t-test and one-way analysis of variance were used to evaluate the differences between groups. Citation management was performed using EndNote X9, Web of Science (https://www.webofscience.com/wos/).

## RESULTS

### Analysis of viral growth kinetics of NDV-Dove strain and its passage

NDV exhibits selective infectivity, being unable to propagate in all cell types, yet capable of infecting specific cells, such as DF-1 and BHK-21 cells (26). The original NDV-Spotted Dove strain and its 100th passage (NDV-Dove[100th]) were infected in DF-1 cells to investigate the mechanisms of cellular infection. NDV-Dove exhibits a preference for infecting DF-1 cells at the single-cell level. The proliferation results indicated that NDV preferentially infects the DF-1 cell line. The increase in these cells facilitates viral replication, which, in turn, results in greater cellular damage. This study involved the infection of DF-1 cells ($1 \times 10^5$ cells/mL) with the original NDV-Dove strain (Dove strain) and the 100th passaged strain (Dove[100th]) at an MOI of 0.01. Supernatants were collected at multiple time points to assess virus titers ($TCID_{50}$/mL) in BHK-21 cells. NDV-Dove was infected with DF-1 cells *in vitro* to assess the changes in expression of the parental NDV-Dove strain at various hours post-infection, in comparison to its 100th passage (NDV-Dove 100th) (Fig. 1A). The passaged strain (NDV-Dove[100th]) induced notable cytopathic effects (CPEs), demonstrating swift alterations in cell growth and presenting a proliferation curve with a virus titer of 4.25 log10 NDV-Dove[100th] at 48 hpi (Fig. 1B). The virus titer diminished as a result of the decreased number of surviving cells relative to the original strain (Fig. 1C). The Dove strain of the virus induces mild syncytium development, which markedly escalates following the 100th passage. This signifies adaptive evolution or the accumulation of mutations during repeated passaging that improves syncytia activity. Enlarged syncytia and plaques indicate enhanced viral cell-to-cell transmission and potentially heightened virulence or fitness of the 100th passage virus. This outcome indicates that the "Dove" virus strain, particularly its 100th passaged strain, promotes substantial cell fusion (syncytium development) in the examined cells (Fig. 1D). The transmitted virus exhibits superior characteristics in inducing this action relative to the original strain, evidenced by the more significant syncytium development and increased syncytial size. The alterations in the virus during passaging, including adaptation to cell culture, may influence its virulence or processes of cell-to-cell transmission.

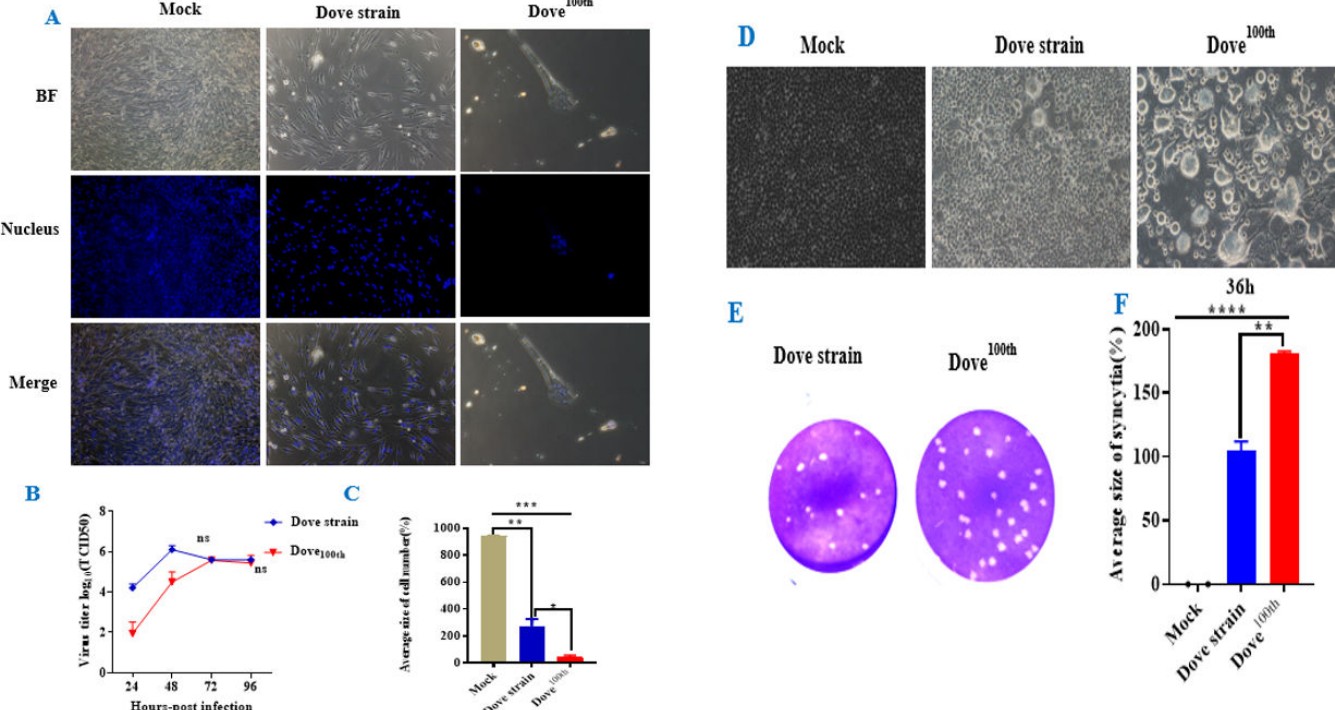

**FIG 1** Comparison of the fusion indices of the virulence of NDV-Spotted Dove strains after the 100th passage. This figure presents the results of the investigation of the effect of viral infection using the "Dove strain" virus and its 100th passage (Dove$^{100th}$) on cell morphology, nuclear integrity, viral replication, and cell viability. (A) Morphological and nuclear changes were induced by the viruses; Dove strain: Parental NDV-Spotted Dove strain, and Dove$^{100th}$, one hundred passages in DF-1 cell (Mock, Dove strain, Dove$^{100th}$) show bright field (BF), DAPI-stained nuclei (Nucleus), and merged images (Merge). The control cell shows a healthy, confluent cell monolayer with intact nuclei and displays the overall cell morphology showing normal cell appearance. Dove strain-infected cells may show morphological changes due to viral infection; cells appear less dense and elongated, with still-visible but slightly affected nuclei. But Dove$^{100th}$ exhibits the most dramatic effect. Cells are sparse and morphologically altered. DAPI staining reveals fewer and more condensed nuclei, suggesting cell death or severe nuclear damage. (B) *In vitro* viral replication kinetics changes in DF-1 cells after viral infection. Cells were infected with each virus at an MOI of 0.01, fixed at 96 hpi. A line graph showing viral titers (log10 TCID$_{50}$/mL) at different time points (24–96 hpi) for Dove strain (blue line) and Dove$^{100th}$ (red line). The parental Dove strain shows increasing viral titers, indicating robust replication due to the morphology of cells; therefore, Dove$^{100th}$ shows initially rising titers but significantly lower than the Dove strain, suggesting reduced replication capacity due to the reduction in cell number. (C) Cell enumeration at 96 hpi, a bar graph showing the average size of cell number (%) for three Mock (gray), Dove strain (blue), and Dove$^{100th}$ (red). The Mock maintains nearly 100% cell number across all three times, Dove strain significantly reduces cell number, indicating CPEs, and Dove$^{100th}$ leads to the most profound reduction in cell number (near 0%), suggesting severe cytotoxicity despite lower replication. Asterisks indicate statistical significance (***$P < 0.001$), confirming that both Dove and Dove100th strains significantly affect cell survival, with Dove100th being the most cytotoxic. Dove strain induces moderate CPE and replicates effectively. Dove 100th, while less competent in replication, is highly cytotoxic, possibly due to accumulated mutations enhancing cell-killing ability. Passaging of the virus from the parental NDV Dove strain to its 100th passage may have selected for a strain with increased pathogenicity but compromised replication efficiency. These findings suggest a potential trade-off between replication and cytotoxicity in viral evolution through serial passaging. (D) The fusogenic activity (syncytium formation) of a viral Dove strain and its 100th passage (Dove$^{100th}$) compared to a mock control infected in the BHK-21 cell at an MOI of 0.001 and fixed at 36 hpi. Mock: Cells show normal morphology, indicating no infection. Cells infected with the Dove strain exhibit some syncytium formation, and we observed some changes in cell morphology compared to the mock, indicating that the virus induces fusion, but in the cell infected with Dove$^{100th}$, markedly increased syncytium formation compared to the original Dove strain, suggesting that repeated passage has enhanced fusogenic capacity. (E) The plaques formed by the two virus preparations are shown: Differences in plaque size, number, and morphology indicated the differences in virus virulence, replication, or cell-to-cell spread in repeated passage. The 100th passage strain plaques are noticeably larger and more diffuse than those from the original Dove strain, indicating more extensive cell fusion and spread. (F) Quantification of syncytium size at 36 h: There is almost no syncytium formation in the control cell, so the average size is near 0%. Cells infected with the original strain cause some syncytium formation, with an average size that is significantly higher than the mock (indicated by the statistical significance, e.g., ****$P < 0.0001$). The passaged virus causes even larger syncytia, as shown by the larger average size compared to the original strain. The statistical significance (asterisks) indicates that these differences are not due to random chance and that the passaged virus has a more potent effect on inducing cell fusion.

## Pathogenicity test analysis

After 100 serial passages of the NDV Dove strain through ECEs, the ICPI of the parental NDV-Dove strain was maintained at 0.521, categorizing it as a lentogenic strain. In contrast, the ICPI of NDV-Dove[100th] rose to 1.43, reclassifying it as mesogenic. The MDT for the parental NDV-Dove strain was 96.1 h, whereas the MDT for the NDV-Dove[100th] strain was 60.9 h (Table 1). The results from the ICPI and MDT assessments indicate that the virulence of the NDV-Dove[100th] strain is greater than that of the parental NDV-Dove strain. Consequently, the IVPI was evaluated in 6-week-old SPF chickens to assess the virulence and replication of both the NDV-Dove strain and its passage counterpart, NDV-Dove[100th]. The IVPI for the parental strain was 0.2, aligning with its lentogenic classification, whereas the IVPI for NDV-Dove[100th] was 1.9, signifying a mesogenic virulence phenotype. Additionally, tissues from the brains, spleens, lungs, proventriculus, and bursae of Fabricius of chickens that died during the ICPI and IVPI tests were collected for histopathological examination (data not shown).

## Determining the effect of different passages of the NDV-Dove strain-infected chickens

To assess the pathological effects and variations in virulence between the original NDV-Dove strain and the passaged NDV-Dove[100th] strain, three chickens from each group were randomly chosen and euthanized at 3 and 5 dpi. Furthermore, histological examinations of pathological sections were conducted on the brain, trachea, lung, spleen, duodenum, and bursa of Fabricius (Fig. 2E). At 1 week post-inoculation (wpi), the chickens that received the parental NDV-Dove strain exhibited no clinical signs, and no fatalities occurred. In contrast, clinical symptoms were noted in the chickens inoculated with the NDV-Dove[100th] strain (Fig. 2A and B). Blood samples were obtained from live chickens on days 7 and 14 following infection and were utilized to assess HI antibody titers (Fig. 2C), in accordance with the guidelines set forth by the OIE. Although *in vitro* replication is lower (as indicated in the previous result, Fig. 1B), Dove[100th] exhibits greater virulence *in vivo*, possibly attributable to immune evasion, modified host interactions, or enhanced tropism. The Dove strain demonstrates safety and immunogenicity, indicating its potential utility in future research, such as vaccine development (Fig. 2C). The serial passage of the original NDV Dove strain through chicken embryos has increased the virulence of the virus. Histological images indicate that the Dove[100th] strain induces tissue-specific alterations in various organs when compared to the Dove strain and PBS control. The Dove[100th] strain typically results in more severe and widespread damage across various tissues (Fig. 2E), suggesting its potentially heightened pathogenicity. The observed histological changes reflect the interaction between the host and pathogen at the tissue level, wherein the virus induces immune responses and results in tissue damage.

TABLE 1    Virulence increased in the NDV-Spotted Dove strain[a]

| Strains | ICPI | MDT | IVPI |
| --- | --- | --- | --- |
| NDV-Spotted Dove | 0.521 | 96.1h | 0.2 |
| NDV-Spotted Dove[100th] | 1.43 | 60.9h | 1.9 |

[a]This table displays virulence indices for two strains of NDV: the original NDV-Spotted Dove strain and its 100th passage, assessed by standard pathogenicity assays. ICPI in 1-day-old SPF chickens infected with a dose of 50 mL per chick, the MDT in 9-day-old chicken embryos, and IVPI in 6-week-old SPF chickens infected with a dose of 100 mL per chicken. The NDV-Spotted Dove (original) exhibited a low ICPI of 0.521 and an IVPI of 0.2, indicating low virulence but has high MDT 96.1 h further corroborates its mild pathogenicity. NDV-Spotted Dove[100th] exhibited a high ICPI (1.43) and IVPI (1.9), indicating a significant increase in virulence and a reduced MDT (60.9 h), confirming a more rapid and severe infection. Serial passaging of NDV in the spotted dove strain results in a significant enhancement of pathogenicity. The rise in ICPI and IVPI, coupled with a diminished MDT, unequivocally signifies that the NDV-Spotted Dove[100th] strain has transformed into a more virulent variety. These findings correspond with prior syncytium evidence (Fig. 1D), indicating that heightened syncytia activity is associated with greater virulence. The 100th passage of NDV-Spotted Dove has markedly more virulence than the parental strain, attributable to adaptive changes that augment viral proliferation and tissue damage. This has significant ramifications for the monitoring and regulation of new NDV mutations.

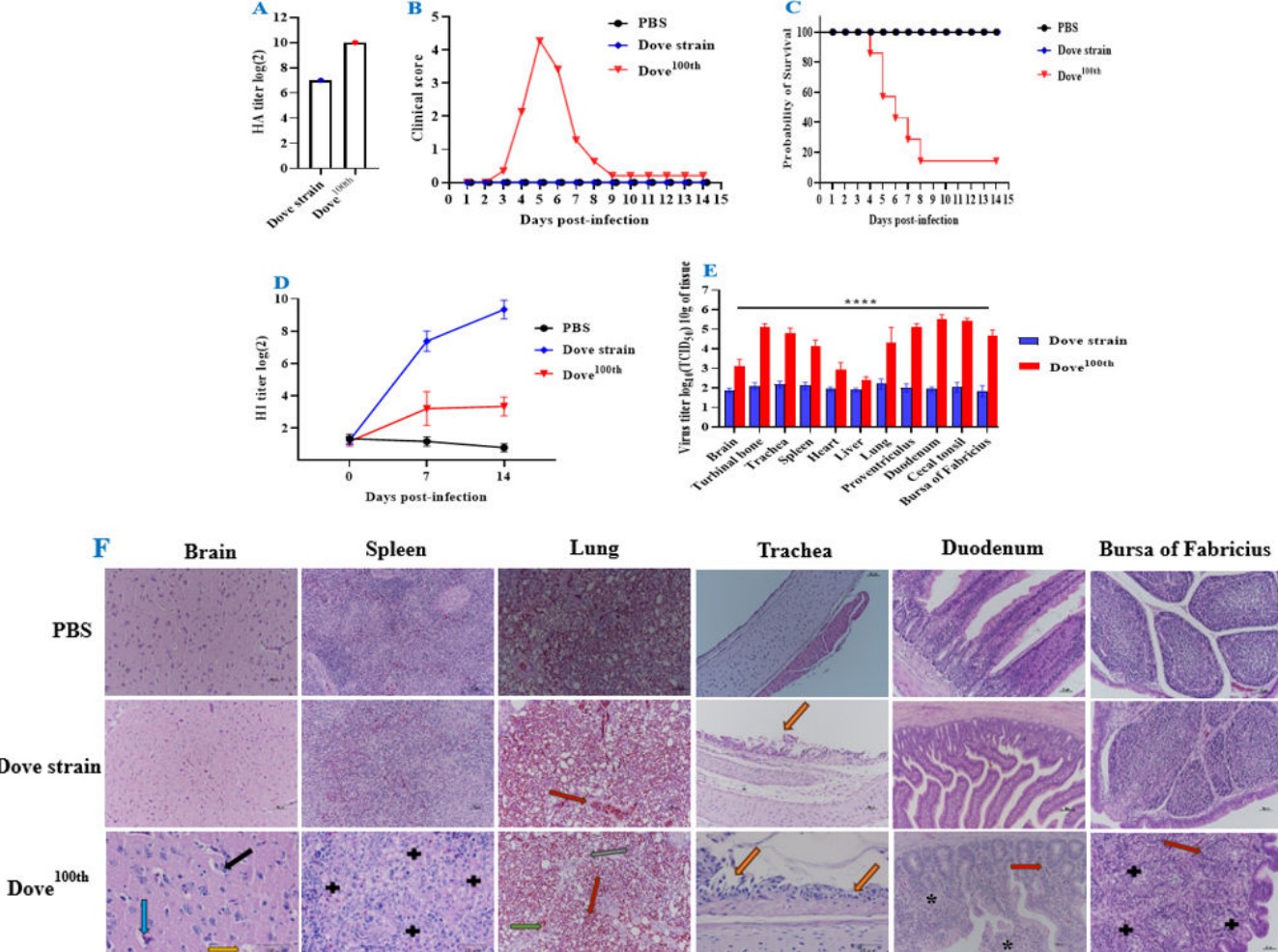

FIG 2 Comparing the pathogenicity, immunogenicity, and tissue tropism of the original Dove strain and its 100th passage (Dove[100th]) in 4-week-old SPF chickens at a dose of $5 \times 10^6$ PFU/mL. Samples were collected from our live chickens at 5 dpi, and virus titers are shown in separate panels. (A) Comparison of HA titers in different Dove Strains: HA titer is related to the ability of hemagglutinin to agglutinate red blood cells. The height of the bars indicates the level of HA titer in each group. The HA titer of the "Dove strain" group is about 7 ($log_2$), and that of the "Dove100th" group is 10 ($log_2$). A higher HA titer may imply a higher content or activity of hemagglutinin. (B) Clinical score progression over time: The line graph illustrates clinical scores recorded over a 14-day period following infection with Dove strain, Dove[100th] groups, and PBS serving as the control group. The clinical score remains near 0 over the 14-day period, indicating that subjects receiving PBS (black) do not display disease-related symptoms. Four-week-old SPF chickens infected with the parental NDV-Spotted Dove strain (blue) initially exhibit a low clinical score. A significant increase in the score is observed, with no clinical symptoms (score = 0), suggesting mild or absent disease. Conversely, Dove[100th] (red) exhibits a pattern akin to the Dove strain, yet may possess distinct kinetics or magnitudes. This present result indicates a peak followed by a decline, demonstrating that the 100[th] Dove strain continues to induce disease symptoms, albeit with potential variations in the disease course relative to the standard Dove strain. Dove[100th] exhibits increased virulence, resulting in a notable elevation in clinical scores that peak between days 5 and 7, signifying severe disease manifestations. (C) Survival Probability: Survival curve over a 14-day period after infection. The PBS group exhibits a survival probability of 100% throughout the 14-day period, indicating no mortality in the control group attributable to the PBS treatment. The Dove strain begins at a survival probability of 100%, but exhibits a decline over time, particularly evident between 5 and 6 dpi. This indicates that infection with the Dove strain results in mortality among the subjects. Consequently, the steep decline in survival observed in Dove100th correlates with increased fatalities among chickens, suggesting varied pathogenic effects that contribute to higher mortality rates. The Dove[100th] strain exhibits lethality, in contrast to the non-lethal nature of the original strain. (D) HI antibody titers were measured (log2) on days 0, 7, and 14 post-infections. The PBS group exhibited no seroconversion, as anticipated; however, the Dove strain elicited a robust, time-dependent antibody response. Dove 100th elicits a limited antibody response, attributable to early mortality or immune suppression. The Dove strain exhibits greater immunogenicity as a vaccine candidate, while Dove100th may evade or suppress the host immune response. (E) The bar graph illustrates viral titers (log10 TCID$_{50}$/mg tissue) across various chicken organs following infection with the parental NDV Dove strain and its 100[th] passage. The Dove[100th] (red) exhibited significantly higher viral loads across nearly all tissues, whereas the Dove strain (blue) demonstrated low or undetectable viral titers in most tissues. Asterisks (****) denote highly significant differences ($P < 0.0001$), indicating that the 100[th] passage (Dove[100th]) enhanced tissue tropism and

Fig 2 (Continued)

systemic dissemination, which correlates with increased pathogenicity, resulting in severe clinical symptoms, mortality, and widespread systemic viral spread. (F) Histopathological analysis of tissues using H&E staining was conducted on 4-week-old SPF chickens infected with PBS (control), Dove strain, and Dove[100th] viruses. Blue arrow indicated edema, black arrow indicated neurophagy, yellow arrow lymphocytic and microglial cells infiltration around the vessel in the brain, black plus (+) showed necrosis in the spleen and bursa of Fabricius, green arrow indicated congestion and gray arrow shrink in the lung, red arrow indicated bleeding, pink arrows showed severe mucosal epithelium necrosis in the trachea, and black asterisks severe mucosal epithelium shedding duodenum, in the bronchi. The evaluated tissues comprise the brain, spleen, lung, trachea, duodenum, and bursa of Fabricius, providing a visual assessment of tissue damage and inflammation. Normal tissues were observed in the PBS control. In contrast, the Dove strain induced only mild or negligible pathology in various tissues. However, neurological damage, severe respiratory and gastrointestinal lesions, lymphoid tissue destruction, and immunosuppression were noted in all tissues of Dove[100th]. The severity of pathology in Dove[100th]-infected chickens is associated with elevated viral titers and previously observed mortality, supporting the conclusion that serial passage resulted in increased virulence and systemic tissue damage.

## Analyses of the overexpression of different amino acids affected the passage of the NDV-Dove strain

After 100 serial passages of the NDV-Dove strain through chicken embryos, 12 distinct mutations were identified in the structural proteins of NDV-Dove[100th] (Fig. 3A). In contrast, the parental NDV-Spotted Dove strain, which showed no mutations, the 100th passage of the virus revealed 12 mutations: two on the P gene (T25M and S369N), one on the M gene (S200N), two on the F gene (S266G and D403D, with no amino acid change), we observed three mutations on the HN gene (N147D, I405L, and K495N). Additionally, four mutations were noted on the L gene (E232E, C875C, A1390A, and F1570F), all of which did not alter the corresponding amino acids. Upon comparing these observations with the sequences found in GenBank KC934170.1, it became evident that the adaptive mutations present in these proteins diverged from the consensus sequences of established NDV strains. The analysis of quasispecies frequency indicated that the Dove[100th]-F mutation, characterized by two amino acid substitutions, did not exhibit a gradual change (Fig. 4E, e i -e iv). In alignment with the study's objectives, six-point mutations were engineered in infectious cDNA clones. pDove served as a template, and specific primers were utilized in an overlap polymerase chain reaction to mutate six individual amino acids at single loci. The mutations T25M, S369N, S200N, N147D, I405L, and K495N were derived from pDove[100th]-P, pDove[100th]-M, and pDove[100th]-HN, respectively. All mutant cDNAs were inserted into the pBR322-Dove vector, which includes designated resistance sites. The plasmids generated were designated as pDove[100th]-$P_{T25M}$, pDove[100th]-$P_{S369N}$, pDove[100th]-$M_{S200N}$, pDove[100th]-$HN_{N147D}$, pDove[100th]-$HN_{I405L}$, and pDove[100th]-$HN_{K495N}$ and were preserved at −20°C.

Plasmids were transfected into BHK cells in conjunction with helper plasmids to facilitate virus rescue. Cell lesions were monitored at 24-h intervals following transfection for a duration of 72 h, with microscopy employed to observe the formation of syncytia. Cell supernatants were harvested at 72 hpt and subsequently inoculated into SPF chicken embryos. The isolated virus underwent dilution and was passaged for five successive passages in 9-day-old SPF chicken embryos. Positive HA titers indicate stable replication of the rescued virus. Following the fifth passage (P5), the detection of stable viral mutations indicated that the rescued virus exhibited significant genetic stability. RNA extraction was conducted, followed by sequencing to verify the mutation sites. The recombinant viruses that were successfully rescued are designated as rDove[100th]-$P_{T25M}$, rDove[100th]-$P_{S369N}$, rDove[100th]-$M_{S200N}$, rDove[100th]-$HN_{N147D}$, rDove[100th]-$HN^{I405L}$, and rDove[100th]-$HN_{K495N}$.

## Growth characteristic analyses of rDove[100th] mutants

The recombinant NDV-Dove[100th] mutants were infected into DF-1 cells *in vitro* at an MOI of 0.01 to evaluate the expression changes of the viruses over time compared to the parental strain. Cell supernatants were collected every 24 hpi to assess virus titers using $TCID_{50}$/mL (Fig. 2B and 3C). At 24 hpi, the replication titers of all recombinant strains displayed comparable lesions to those of the parental rDove strain. Replication titers

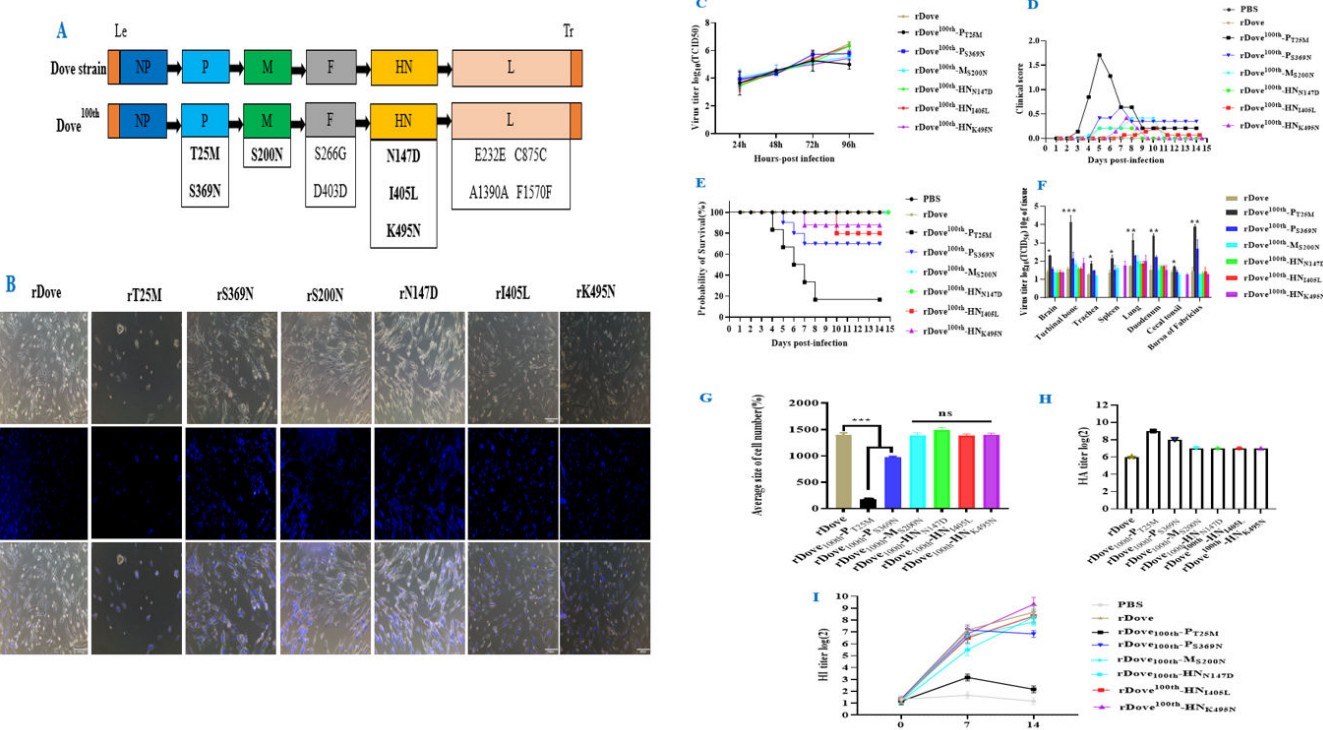

FIG 3 The genetic evolution and mutations that occur during the passage of the original Dove strain through chicken embryos. (A) The schematic illustrated the variability of each substitution throughout the passage, contrasting the Dove strain with its 100th passage strain. Mapping the mutations in essential viral proteins (NP, P, M, F, HN, and L) is conducted. This information aids in comprehending viral adaptation, changes in pathogenicity, and the potential development of vaccines or antivirals, as mutations in these proteins can influence viral function, infectivity, and antigenicity. Identified specific amino acid mutations in the 100th passage are enumerated under the corresponding gene: P (T25M, S369N), M (S200N), F (S266G, D403D); HN: N147D, I405L, K495N; L: E232E, C875C, A1390A, F1570F. Mutations such as D403D, E232E, C875C, A1390A, and F1570F are synonymous, indicating no change in amino acid sequence. In contrast, other mutations are nonsynonymous, resulting in an amino acid change that may influence protein function. Amino acid substitutions (e.g., T25M, S369N) reflect differences between the two strains at specific positions within the proteins, potentially influencing protein function, antigenicity, or other biological properties. (B) Functional characterization of mutants: This figure examines the effects of specific mutations on the viral phenotype through immunofluorescence microscopy. Images of DF-1 cells infected with the recombinant Dove strain and the recombinant Dove$^{100th}$ mutants were observed. rDove exhibits significant fluorescence during viral replication, while mutants (rT25M, rS369N) demonstrate diminished fluorescence, indicating that these mutations hinder viral replication. The mutations (rN147D, rI405L, rK495N, rS200N) exhibit fluorescence comparable to rDove, suggesting that these alterations may not substantially impact function. This figure indicates that certain mutations acquired during viral passaging, particularly in P, hinder viral replication, whereas mutations in M, F, HN, and L do not exhibit this effect. They may be pertinent for comprehending viral attenuation and for the advancement of vaccines. (C) The figure further analyzes viral mutants obtained from the 100th passage of the dove strain, evaluating their impact on viral replication, pathogenicity, and immunological or cellular outcomes through various in vitro methods. Virus titer (log$_{10}$ TCID$_{50}$/mL) indicates that all mutants, including rDove and its derivatives, demonstrate comparable replication kinetics in vitro. (D) Clinical score: The clinical score ranges from 0.0 to 2.0. Each line represents a distinct treatment or control group. The PBS group exhibits a consistently low and stable clinical score over time, suggesting that chickens receiving only PBS do not manifest significant clinical symptoms associated with the experimental infection. Certain rDove variant groups, including rDove$^{100th}$-P$_{T25M}$, exhibit a peak in clinical scores approximately 4 to 5 dpi, subsequently followed by a decline. This indicates that the T25M mutant variants initially lead to an increase in clinical symptoms, which subsequently diminish over time. Other rDove variant groups exhibit lower and more stable clinical scores during the observation period, suggesting that these genetic modifications may have diminished the pathogenicity or the capacity of the virus to induce clinical symptoms in chickens. (E) Survival probability (%): The PBS, rDove, rDove$^{100th}$-M$_{S200N}$, and rDove$^{100th}$-HN$_{N147D}$ groups exhibited a 100% survival rate over the 14-day observation period. The findings suggest that chicken in both the control (PBS) and the specific rDove variants (rDove, rDove$^{100th}$-M$_{S200N}$, and rDove$^{100th}$-HN$_{N147D}$) did not experience mortality due to the experimental conditions. Other rDove variant groups, such as rDove$^{100th}$-P$_{T25M}$, begin with a survival rate of 100% but exhibit a swift decrease in survival probability within the initial days following infection. Certain groups exhibit a low survival probability, approximately 20%, by the conclusion of the observation period. Groups such as rDove$^{100th}$-P$_{S369N}$ and rDove$^{100th}$-HN$_{K495N}$ exhibit intermediate survival patterns, characterized by a gradual decline in survival probability over time, though less pronounced than that observed in certain other variants. (F) The virus titer (log$_{10}$ TCID$_{50}$/g of tissue) across various tissues in infected chickens. rDove functions as a reference recombinant Dove virus group for comparative analysis. rDove$^{100th}$-P$_{T25M}$, rDove$^{100th}$-P$_{S369N}$, rDove$^{100th}$-M$_{S200N}$, rDove$^{100th}$-HN$_{N147D}$, rDove$^{100th}$-HN$_{I405L}$, and rDove$^{100th}$-HN$_{K495N}$. Genetically modified variants of the rDove virus, characterized by specific

**Fig 3 (Continued)**

genetic alterations through passaging, are being investigated for their effects on viral replication across various tissues. The rDove$^{100th}$-P$_{T25M}$ variant in brain tissue exhibits a significantly elevated virus titer, indicating a highly significant difference. Other variants exhibit differing levels, with some displaying notable differences in comparison to rDove. In tissues such as the turbinat bone, trachea, and lung, various variants display distinct virus titers, with certain variants significantly differing from others. In lung tissue, multiple variants exhibit elevated titers with significant differences. In lymphoid tissues, including the spleen and Bursa of Fabricius, as well as intestinal-related tissues such as the duodenum and cecal tonsil, variations in virus titers among the variants are observed, with rDove$^{100th}$-P$_{T25M}$ exhibiting significant increases relative to the rDove group. Mutants, particularly those with P mutations (T25M and S369N), exhibit significantly elevated counts ($P < 0.001$, denoted as ***), a significant difference ($P < 0.01$, denoted as **), and a statistically significant difference ($P < 0.05$). These notations facilitate the identification of virus variants that exhibit significantly different virus titers in specific tissues (T25M). (G) The average cell size percentage shows that the groups marked with asterisks (rDove, rDove$^{100th}$-P$_{T25M}$, and rDove$^{100th}$-P$_{S369N}$) exhibit a highly significant difference ($P < 0.001$) when compared to other relevant groups. This indicates that the factor, exemplified by the specific genetic modification rDove$^{100th}$-P$_{T25M}$, significantly influences the average cell number percentage. This significantly alters the proliferation, aggregation, or size distribution characteristics of cells, leading to a notable difference in this statistic. The M (S200N) and HN (N417D, I465L, K495N) mutations assessed do not significantly impact fusogenic activity, suggesting that these residues are not essential for fusion regulation in the given conditions. (H) Recombinant Dove virus HA titer analysis: This HA titer figure provides valuable insights into the hemagglutination properties of a panel of recombinant Dove virus strains. The significant variation in HA titers among strains reflects differences in viral hemagglutinin function, which, in turn, has implications for host virus interactions, immune responses, and viral evolution. The lower recombinant HA activity (MS200N and HN [N417D, I465L, K495N]) may indicate reduced fitness in natural host infection, potential attenuation (useful for vaccine development), or altered receptor specificity. But the rDove100th-PT25M mutation that enhances virulence during the passage has the ability to bind to red blood cell receptors, increasing the HA titer. (I) Recombinant Dove virus HI titer analysis: Some modified virus groups (MS200N and HN [N417D, I465L, K495N]) show similar trends to the original rDove group but with different titer levels. They show a higher HI titer at day 14 compared to the original rDove group. It may indicate that the mutation in this strain enhances its ability to induce an anti-HI antibody response. This could be related to changes in viral antigenicity, making it easier for the immune system to recognize and elicit a stronger antibody response. These are groups where specific mutations (such as PT25M, PS369N mutations in different viral proteins) have been introduced into the rDove virus strain. By comparing their HI titer curves with the original rDove group and the PBS group, we can analyze the impact of these mutations on the induced immune response, especially to rDove100th-PT25M. This may suggest that the PT25M mutation, which has relatively lower HI titers at later time points, may affect the virus's ability to induce a robust anti-HI antibody response, perhaps by altering the structure of viral antigens involved in antibody recognition, leading to a weaker immune stimulation effect.

for rDove$^{100th}$-HN$_{N147D}$, rDove$^{100th}$-HN$_{I405L}$, and rDove$^{100th}$-HN$_{K495N}$ increased consistently until 96 hpi. Between 48 and 72 hpi, the replication capacity of rDove$^{100th}$-P$_{T25M}$ showed a significant decline. From 72 hpi to 96 hpi, virus titers decreased markedly due to a reduced number of surviving cells in comparison to the parental rDove strain. In contrast, rDove$^{100th}$-P$_{S369N}$ displayed a comparable lesion pattern, whereas rDove$^{100th}$-M$_{S200N}$ demonstrated lesions akin to those of the parental strain over the duration of the study. The membrane fusion index for rDove$^{100th}$-P$_{T25M}$ exhibited the most significant effect, producing the largest syncytial effects on cell lesions, which displayed greater variability compared to the other mutant strains (Fig. 3B) and in relation to the parental recombinant (rDove). The findings demonstrate that the T25M point mutation in the P gene of rDove$^{100th}$ significantly impacts the virus's proliferation capacity *in vitro*. Analysis of genetic variations between the Dove strain and its 10th passage strain. These substitutions serve as direct indicators of genetic variation at the protein-coding level. The specific positions of these changes may influence the structure and function of the proteins. Amino acids possess unique chemical properties. In the context of viral proteins, such alterations may influence functions such as viral attachment. Analyzing these potential functional changes aids in forecasting the behavioral differences of the 100th passage strain relative to the original Dove strain, particularly regarding infectivity, host range, and virulence. Mutations in the Dove strain can exhibit diverse impacts on virulence, as evidenced by mortality rates (Table 2). Certain mutations enhance virulence (e.g., rDove$^{100th}$-P$_{T25M}$, rDove$^{100th}$-P$_{S369N}$), while others exhibit minimal impact (e.g., rDove$^{100th}$-M$_{S200N}$, rDove$^{100th}$-HN$_{N147D}$). Additionally, some mutations may result in delayed or restricted mortality (e.g., rDove$^{100th}$-HN$_{I405L}$, rDove$^{100th}$-HN$_{K495N}$). These results enhance our understanding of the molecular basis of viral virulence and facilitate the identification of key viral protein regions associated with pathogenicity.

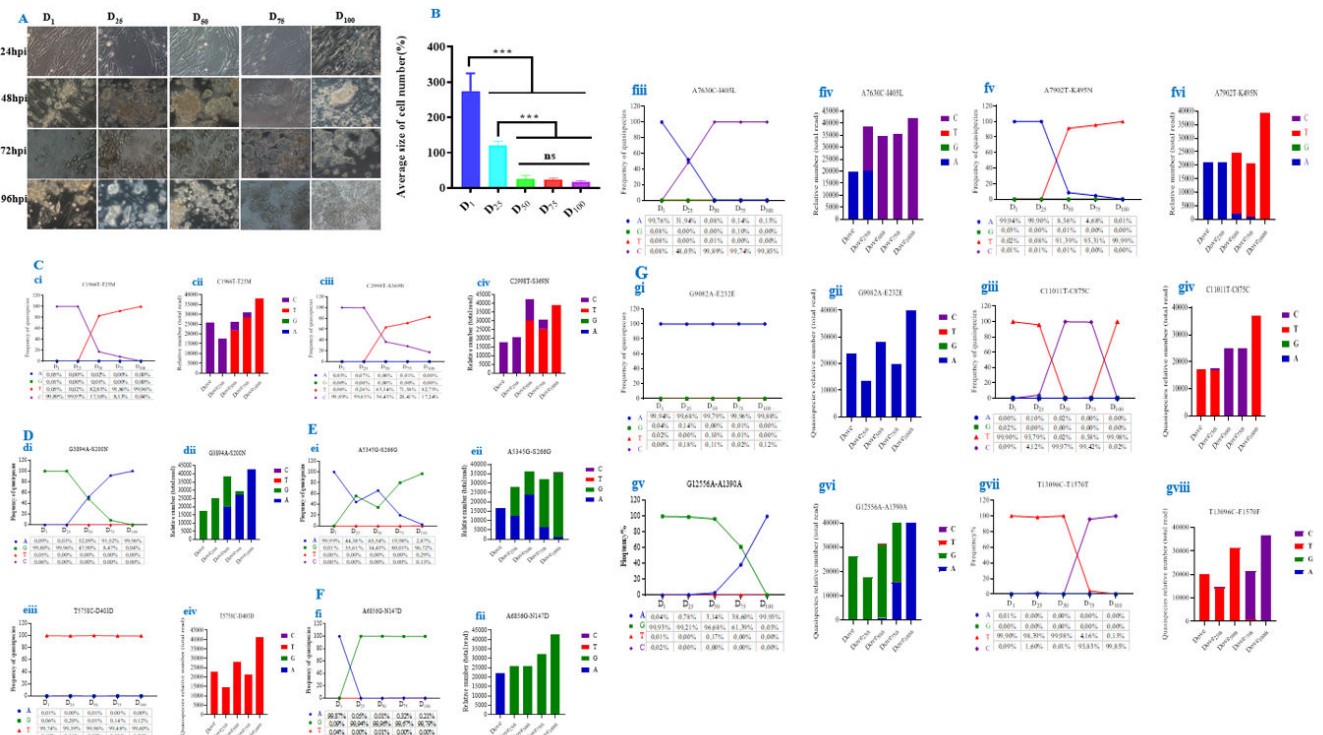

**FIG 4** Proportion and mutation frequency quasispecies target enhanced virulence during different passages of NDV-Dove strain. (A) Microscopy images represent different cell lines: Cells were infected with each virus at 0.01 of MOI, fixed at 96 hpi. D1, D25, D50, D75, and D100 represent the original Dove strain, 25th, 50th, 75th, and 100th passage, respectively. By observing the images, one can assess the CPE of the virus on the cells over time. From 24 hpi, the cells may appear relatively normal, but as time progresses (e.g., by 96 hpi), there could be signs of cell rounding, detachment, and syncytia formation, which are typical CPEs. Comparing across the different $D_1$ and $D_{100}$ can show which conditions lead to more or less severe CPE. $D_{100}$ has a stronger impact on the cells. (B) Average size of the cell number: The average size of the cell number, expressed as a percentage, measures the degree of cell growth, aggregation, and related characteristics of the cell population. $D_1$ has the largest average size of cell number, significantly higher than the other groups. This suggests that the original Dove strain has a greater quantity or a characteristic related to larger "size" in terms of the measured index. The 25th passage presented a lower average size of cell number than that of the original Dove strain but higher than the 50th, 75th, and 100th passage strains, which have relatively low average sizes of cell number, and the difference between them is not significant (marked as "ns"), while they are significantly different with $P$ values ($P < 0.001$) from the original Dove strain to its passaged strains. (C) The original Dove strain and its passaged strains (D1, D25, D50, D75, D100), by measuring how often different nucleotide variants appear and how many times they are read, these mutations change as the virus goes through different stages. We evaluated viral genetic stability, adaptation mechanisms, and the emergence of dominant viral variants, which are crucial for understanding viral pathogenesis and transmission and for developing effective control strategies. C1966T - T25M mutation (C, c i ): At the initial passage (D1), the nucleotide C dominates with a frequency of 99.89%. As passages progress, the frequency of C decreases sharply, while the frequency of T increases. By D100, T has a frequency of 99.96%, indicating that the C1966T mutation becomes the dominant variant over time. The minor nucleotides A and G remain at very low frequencies throughout. C2998T-S369N mutation (C, c iii): At D1, C is the predominant nucleotide (99.89%), similar to the previous mutation. Over successive passages, the frequency of C drops, and T rises. By D100, T reaches 82.75%, indicating that the C2998T mutation becomes dominant in the viral population during these passages and G maintain negligible frequencies. C1966T - T25M Mutation (C, c ii ): The bar graphs show the relative number of reads for different nucleotides. In the initial Dove sample, the nucleotide C has a higher relative read number. As passages progress, the relative read number of T increases significantly, especially in Dove100th, where it far exceeds that of C. This mirrors of the frequency data shows the dominance shift at the sequence read level. C2998T - S369N Mutation (C, c iv): Initially, C has a higher relative read number in the Dove sample. As the passages continue, the relative read number of T goes up, and in Dove100th, it is the most noticeable, showing that the mutation is becoming dominant and taking over the original nucleotide in the viral population. The present result clearly demonstrates that for both the C1966T-T25M and C2998T-S369N mutations in P gene, there are a shift from the initial nucleotide (C) to the mutated nucleotide (T) over successive viral passages. The frequency of the mutated nucleotide increases steadily, and by the 100[th] passage, it becomes the dominant variant in the viral population. The relative read number data further validate this shift at the sequencing read level. The evidence indicates that these mutations confer a selective advantage to the virus during the passage process, likely affecting viral fitness, replication, or interaction with host factors. Understanding these mutation dynamics is essential for predicting viral evolution and for developing strategies to target these evolving viral variants. For the specific nucleotide positions C1966T (associated with T25M amino acid change) and C2998T (associated with S369N amino acid change), there is a clear trend of the mutated nucleotide (T) replacing the original nucleotide (C) over successive passages. In both cases, starting from a very low frequency of the mutated form at the initial passage (D1), the frequency of T increases (Continued on next page)

Fig 4 (Continued)

steadily, reaching a dominant position by the 100th passage (D100). This indicates that these mutations are being fixed in the viral population, suggesting they confer a selective advantage. The fact that the mutations become dominant implies that they likely enhance the virus's ability to replicate, survive, or interact with the host environment. For example, the amino acid changes (T25M and S369N) might alter the structure or function of the relevant viral proteins (P in these cases) in a way that is favorable for the virus. This could involve improved binding to host cell receptors, more efficient replication processes, or evasion of host immune responses. The consistent shift towards the mutated forms over multiple passages shows that the virus is evolving in a directional manner at these specific genetic loci. It is not a random fluctuation of genetic variants but a clear progression towards a particular genetic state. This directional evolution can help in predicting future genetic changes if the virus continues to be passaged under similar conditions. The initial presence of multiple quasispecies (different nucleotide variants) at low frequencies and the subsequent dominance of one variant (T) highlight the quasispecies nature of the virus. The viral population is a heterogeneous mix of variants, and selective pressures act to drive the dominance of certain variants over others. This understanding of quasispecies dynamics is crucial for grasping how the virus can adapt rapidly to new environments or host defenses. Similar interpretations apply to mutations M (D, d ⅰ -d ⅱ ), F (E, e ⅰ -e ⅱ ), HN (F, f ⅰ -f ⅵ ), and L (G, g ⅰ -g ⅷ ).

## Virulence of different recombinant viruses

Mutations in NDV-Dove[100th] were rescued to identify the amino acid positions that enhance the NDV-Dove strain after 100 serial passages through chicken embryos. Pathogenicity indices, such as MDT and ICPI, were employed to evaluate the virulence of NDV-Dove and its recombinant mutants. Chicks were inoculated with freshly prepared allantoic fluid, diluted 10-fold in PBS, with 50 µL administered to each bird. The results presented in Table 3 indicate that the strains rDove[100th]-$P_{T25M}$, rDove[100th]-$P_{S369N}$, rDove[100th]-$M_{S200N}$, rDove[100th]-$HN_{N147D}$, rDove[100th]-$HN_{I405L}$, and rDove[100th]-$HN_{K495N}$ exhibited ICPI values of 1.37, 1.03, 0.85, 0.46, 0.43, and 0.45, respectively, whereas the parental rDove strain recorded an ICPI of 0.68. The rDove[100th]-$P_{T25M}$ strain demonstrated significantly greater virulence, evidenced by an ICPI of 1.37, compared to the other recombinant strains. MDT values were recorded (Table 3), with rDove[100th]-$P_{T25M}$

**TABLE 2** Mortality rates observed in the animal experiments (recombinant strain)[a]

| Factor | Impact on mortality rates | | Total |
|---|---|---|---|
| | Day 7 | Day 14 | |
| PBS | 0% | 0% | 0% |
| rDove strain | 0% | 0% | 0% |
| rDove[100th]-$P_{T25M}$ | 40% | 20% | 60% |
| rDove[100th]-$P_{S369N}$ | 30% | 0% | 30% |
| rDove[100th]-$M_{S200N}$ | 0% | 0% | 0% |
| rDove[100th]-$HN_{N147D}$ | 0% | 0% | 0% |
| rDove[100th]-$HN_{I405L}$ | 0% | 20% | 20% |
| rDove[100th]-$HN_{K495N}$ | 10% | 0% | 10% |
| Infection route | Oculonasal | | |

[a]This table demonstrates data about the effects of various viral strains (mutants of a passaged dove) on the mortality rates of test subjects (chickens) when infected through the oculonasal route. PBS was used as a negative control: The absence of mortality at Day 7, Day 14, and overall indicates that the buffer is non-pathogenic and does not induce fatalities in the test system. The rDove strain also functions as a control. A 0% mortality rate shows that this strain does not induce mortality in chickens within the 14-day observation period, and the virus is classified as avirulent. rDove[100th]-$P_{T25M}$ has a 40% mortality rate at Day 7 and 20% at Day 14, resulting in an overall mortality rate of 60%. This suggests that the mutant strain is pathogenic. The high mortality (Day 7) indicates a swift pathogenic mechanism, and the persistent mortality by Day 14 results in a considerable overall mortality rate. The T25M mutation may correlate with heightened pathogenicity. rDove[100th]-$P_{S369N}$ exhibits a 30% mortality rate at Day 7 and 0% at Day 14, resulting in an overall mortality rate of 30%. Mortality is predominantly observed at Day 7. The mutation at position S369N may enhance virulence; nevertheless, the host response could potentially regulate the infection by Day 14, leading to no further death. rDove[100th]-$M_{S200N}$ exhibits 0% mortality at Day 7, Day 14, and overall. This indicates that the mutation at position S200M in this recombinant, passaged viral strain did not produce a pathogenic phenotype in the experimental setting. The mutation may not influence critical virulence factors or could potentially be detrimental to the virus. rDove[100th]-$HN_{N147D}$, with a mortality rate of 0% at all time periods, the mutation at position 147 in the HN did not result in enhanced virulence. rDove[100th]-$HN_{I405L}$ exhibits 0% mortality at Day 7, escalating to 20% by Day 14, resulting in an overall mortality rate of 20%. The delayed mortality seen alone on Day 14 indicates that the mutation at position 405 in the HN protein may lead to a more gradual pathogenic mechanism. The virus may multiply or inflict damage progressively, resulting in fatality at a subsequent time point. rDove[100th]-$HN_{K495N}$ exhibits a mortality rate of 10% at Day 7 and 0% at Day 14, resulting in an overall mortality rate of 10%. The mutation at position 495 in the HN protein results in reduced virulence, allowing the host to potentially mitigate further death by Day 14.

**TABLE 3** Virulence of different mutations viruses[a]

| Strains | ICPI | MDT |
|---|---|---|
| rDove | 0.68 | 92.2 h |
| rDove$^{100th}$-P$_{T25M}$ | 1.37* | 70.4 h |
| rDove$^{100th}$-P$_{S369N}$ | 1.03 | 72 h |
| rDove$^{100th}$-M$_{S200N}$ | 0.85 | 72 h |
| rDove$^{100th}$-HN$_{N147D}$ | 0.46 | 99.4 h |
| rDove$^{100th}$-HN$_{I405L}$ | 0.43 | 97 h |
| rDove$^{100th}$-HN$_{K495N}$ | 0.45 | 95.4 h |

[a]This table provides data on various recombinants of different mutations of the Dove strain. The ICPI was assessed in 1-day-old SPF chicks infected with a dose of 50 mL per chick, alongside the MDT evaluated in 9-day-old chicken embryos. rDove exhibits an ICPI value of 0.68 and an MDT of 92.2 h. This suggests reduced virulence. The rDove$^{100th}$-P$_{T25M}$ strain exhibits a high ICPI value of 1.37, indicated by an asterisk, suggesting potential statistical significance in comparison to other strains. The mean duration until death (MDT) is 70.4 h, indicating that infected hosts exhibit a faster mortality rate relative to the rDove strain. rDove$^{100th}$-P$_{S369N}$ exhibits an ICPI of 1.03 and an MDT of 72 h, indicating a relatively high virulence, with a MDT shorter than that of rDove. rDove$^{100th}$-M$_{S200N}$ exhibits an ICPI of 0.85 and an MDT of 72 h, suggesting moderate virulence similar to rDove$^{100th}$-P$_{S369N}$ in terms of MDT. rDove$^{100th}$-HN$_{N147D}$ has an ICPI of 0.46 and an MDT of 99.4 h; rDove$^{100th}$-HN$_{I405L}$ has an ICPI of 0.43 and an MDT of 97 h; rDove$^{100th}$-HN$_{K495N}$ has an ICPI of 0.45 and an MDT of 95.4 h. These strains exhibit lower virulence compared to others, as evidenced by their lower ICPI and relatively low virulence in chickens. Strains exhibiting elevated ICPI values are associated with reduced MDT, indicating a positive correlation between virulence, as quantified by ICPI, and the rapidity of host mortality. The strain marked with an asterisk (rDove$^{100th}$-P$_{T25M}$) appears to be an outlier regarding virulence compared to the other strains.

exhibiting the shortest MDT at 70.4 h. This was followed by rDove$^{100th}$-P$_{S369N}$ and rDove$^{100th}$-M$_{S200N}$, both at 72 h, rDove$^{100th}$-HN$_{N147D}$ at 99.4 h, rDove$^{100th}$-HN$_{I405L}$ at 97 h, and rDove$^{100th}$-HN$_{K495N}$ at 95.4 h. The parental rDove strain demonstrated an MDT of 92.2 h. The findings indicate that the T25M mutation in the P gene and the S369N mutation have a substantial impact on viral virulence, with rDove$^{100th}$-P$_{T25M}$ exhibiting the highest level of virulence. The mutations at amino acid positions 200, 147, 405, and 495 exhibit some influence on virulence. The amino acid mutation at position T25M, especially in the P gene, significantly impacted viral virulence.

## Identification of point mutation and analysis of the key gene affecting the passage of the NDV-Dove strain

To confirm the key sites influencing virulence during the 100th serial passage, ICPI and MDT assays (Table 3) were employed to evaluate the differences in virulence between the parental strain (rDove) and its recombinant variants: rDove$^{100th}$-P$_{T25M}$, rDove$^{100th}$-P$_{S369N}$, rDove$^{100th}$-M$_{S200N}$, rDove$^{100th}$-HN$_{N147D}$, rDove$^{100th}$-HN$_{I405L}$, and rDove$^{100th}$-HN$_{K495N}$. Three-week-old SPF chicks were inoculated via eye drop and nasal drop with the viruses. At 5 dpi, three chickens from each group were randomly selected and euthanized. Tissues such as the brain, turbinate, trachea, lung, cecum, duodenum, proventriculus, cecal tonsil, liver, spleen, and bursa of Fabricius were collected for histopathological examination. However, infected chicken tissues with rDove$^{100th}$-P$_{T25M}$ demonstrated markedly elevated replication levels relative to the parental rDove strain ($P<0.001$). Figure 3F and G illustrates that four chickens infected with rDove$^{100th}$-P$_{T25M}$ exhibited clinical signs of NDV on day 3 post-infection (dpi). By day 4, two chickens exhibited mild signs of mental depression, one displayed drooping wings and mouth breathing, and another died. At 5 dpi, one chicken deteriorated to a prostrate state, nearing death, while four additional chickens showed signs of depression (6/10), and another chicken has died. At 6 dpi, three chickens exhibited signs of depression, while one was prostrate and near death; one chicken succumbed. At 7 dpi, three chickens exhibited signs of depression (6/10), and one chicken succumbed, while at 8 dpi, an additional death was recorded, resulting in a cumulative total of five deceased chickens. No new deaths occurred between days 9 and 14 dpi; two chickens remained normal, while three chickens exhibited signs of depression until 14 dpi. The findings indicate that adaptive mutations in rDove$^{100th}$-P$_{T25M}$ enhanced replication efficiency in chickens.

## Recombination and reassortment of the virus

The viral fitness and quasispecies of both rDove and rDove[100th]-$P_{T25M}$ viruses were evaluated through 10 passages in SPF chicken embryos, utilizing three distinct dilutions of the viral combination (1:1, 1:9, and 9:1). The first, fifth, and tenth passages were cloned into the pMD[TM]19-T vector utilizing particular primers to evaluate viral fitness and reassortment (27). The present study results showed that during the first passage, the mixture in the 1:1, 1:9, and 9:1 dilution exhibited a specific distribution of C (rDove) and T (rDove[100th]-$P_{T25M}$) viruses similar to the 5th passage. However, after 10 serial passages, the virus mixture consisting of rDove (C) and rDove[100th]-$P_{T25M}$ (T) contained 100%T and 0% C, regardless of the dilution ratio (Fig. 6C). Moreover, when rDove and rDove[100th]-$P_{T25M}$ were used to determine virus titers in chicken embryos, rDove[100th]-$P_{T25M}$ yielded significantly higher $TCID_{50}$/mL titers compared to rDove (Fig. 6B).

## Viral fitness competition under green and red fluorescence proteins

DF-1 cells underwent co-infection with rDove-EGFP and rDove[100th]-$P_{T25M}$-mCherry isolates at equal ratios of 1:1, 1:9, and 9:1, utilizing an MOI of 0.01. Supernatants were collected at 48 hpi, adhering to the procedure outlined for the virus growth curves at 5 dpi. Throughout the observations, from the first to the fifth passage, red fluorescence increasingly surpassed green fluorescence across all dilution ratios. Green fluorescence was observed at the sixth passage in the 1:1 dilution ratio. In the 1:9 ratios, red fluorescence predominated over green fluorescence. In the 9:1 ratio, green fluorescence remained detectable, albeit at a reduced intensity. Between the seventh and tenth passages, red fluorescence became entirely dominant, accounting for 100% of the fluorescence signal in the cells (Fig. 7B). The results demonstrate that rDove[100th]-$P_{T25M}$ significantly affects the passage and replication dynamics of the NDV-Dove strain over 100 consecutive passages in chicken embryos.

## DISCUSSION

This study investigated to assess the persistence of immunogenicity during multiple passages and evaluated the protective efficacy of both the original Dove strain and its 100th passaged strains against virulent NDV (vNDV). The parental NDV Spotted-Dove strain was serially propagated 100 times in 9-day-old SPF chicken embryos. Recently, the study has investigated the transmission of NDV in chicken embryos (28, 29). After 100 serial passages in ECE, the parental NDV-Spotted Dove strain and its 100th passage strain (Dove[100th]) were inoculated into $1 \times 10^5$ DF-1 cells at an MOI of 0.01. Observations of viral growth dynamics indicated that the 100th passaged strain (Dove[100th]) produced syncytia earlier (30) than the parental Dove strain in DF-1 cells, similar to BHK-21 cells, indicating accelerated viral replication (Fig. 1A through D and 5A). This research evaluated the virulence of the NDV-Dove strain through the implementation of standard virulence assessments, including the ICPI (31). The results compared the passaged (Dove[100th]) strain to the original lentogenic NDV-Spotted Dove strain, which exhibited an ICPI of 0.425. The passaged strain demonstrated markedly elevated ICPI, MDT, and IVPI values, signifying enhanced virulence relative to the parental strain (Table 1). After 100 serial passages through chicken embryos, the NDV-Dove strain evolved into a mesogenic phenotype, indicating increased virulence (20), while the parental NDV-Dove strain remained lentogenic virulence. The virulence of the NDV-Dove strain could change after serial passage in chicken embryos. It may evolve from a low-virulence strain to a high-virulence strain, increasing its pathogenicity in chickens. Low-virulence NDV strains isolated from ducks may grow into highly virulent variants following continuous passage in chicken air sacs. Nonetheless, not all pathways will result in enhanced virulence. Research indicates that following 20 serial passages of a duck-origin lentogenic isolate in chicken embryos, the strain retains its lentogenic properties (32). The evolution of the NDV-Dove strain following serial passage in chicken embryos is a multifaceted process influenced by various factors. This evolutionary route and traits are contingent upon the genetic alterations in the virus and the dynamics of its interaction with the host. The

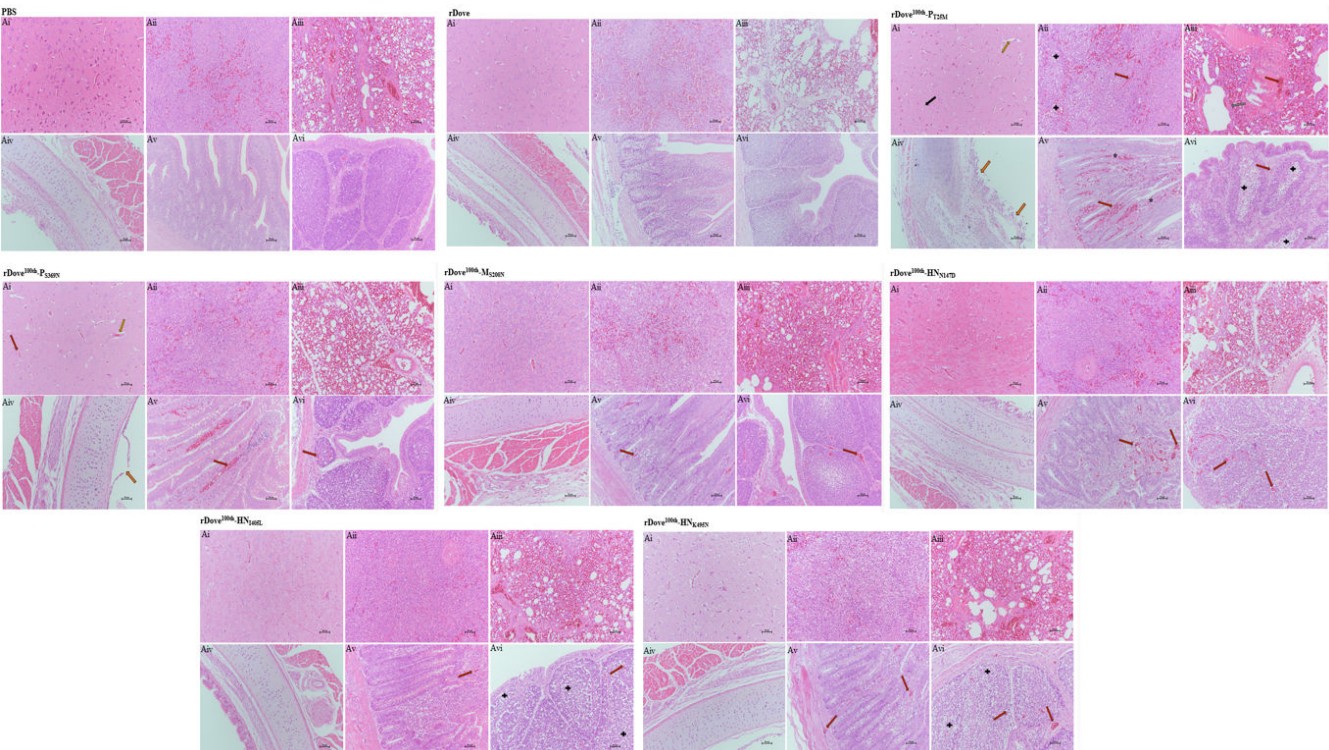

**FIG 5** Histological staining (hematoxylin and eosin) in 3-week-old SPF chickens infected with six different mutations of viruses at a dose of $5 \times 10^6$ PFU/mL: (A i ) brain, (A ii ) spleen, (A iii ) lung, (A iv ) trachea, (A v ) duodenum, and (A vi ) Borsa of Fabricius (hematoxylin and eosin). Virus titers in 3-week-old chickens inoculated with six different mutations of viruses during the passage of NDV-Dove strain, infection dose was $5 \times 10^6$ PFU/mL, and samples were collected in outlive chickens at 5 dpi. Blue arrow indicated edema, black arrow indicated neurophagy, yellow arrow lymphocytic and microglial cells infiltration around the vessel in the brain, black plus (+) showed necrosis in the spleen and bursa of Fabricius, green arrow indicated congestion and gray arrow shrink in the lung, red arrow indicated bleeding, pink arrows showed severe mucosal epithelium necrosis in the trachea, black asterisks severe mucosal epithelium shedding duodenum, in the bronchi. Collectively, these slides represent normal histological appearances of various tissues in an organism treated with PBS (a control treatment). There are no obvious signs of disease, inflammation, necrosis, or abnormal cell growth, suggesting that the PBS treatment does not induce pathological changes in these tissues and served as a baseline for comparison with other experimental groups. The rDove group shows negligible pathology in various tissues. Similar to the PBS group, there are no evident signs of disease, inflammation, abnormal cell proliferation, or tissue damage. This suggests that this does not induce pathological changes in these tissues, and these slides can also serve as a baseline for comparison with other experimental studies to identify any treatment-associated impacts on tissue structure. The rDove[100th]-P[T25M] variant appears to have a pathogenic effect on the tissues examined. It causes various degrees of cellular damage, inflammation, and disruption of normal tissue architecture across different tissue types. These findings suggest that this specific viral construct induced significant pathological changes in the host tissues, which could contribute to its overall virulence. The rDove[100th]-M[S200N] variant appears to have a pathogenic effect on the examined tissues. It causes a variety of changes, including cellular stress, inflammation, disruption of tissue architecture, and damage to epithelial layers. The rDove[100th]-HN[N147D], rDove[100th]-HN[I405L], and rDove[100th]-HN[K495N] variants appear to have a similar pathogenic effect on the examined tissues. They caused various pathological changes, including cellular stress and inflammation. These findings suggest that these viral constructs cannot induce a significant tissue-level alteration, which may not contribute to their overall virulence.

substantial quantity of viral progeny generated during passage indicates the selective pressures exerted within the host (33).

In an experiment aimed at examining and regulating the humoral immune response, excretion, and transmission of pathogenic NDV in chickens, NDV was primarily transmitted via indirect contact with infected pigeons or birds, which subsequently facilitated indirect transmission to chickens. A distinct study indicated that NDV evolves in response to its environment (34). The direct contact of the virus with an experimental model yields insights into the evolution of the infection and the dissemination of the virus, encompassing the emergence of clinical symptoms, strategies for prevention, and the mitigation of viral excretion. In the animal experiment, 4-week-old SPF chickens were orally infected with either the parental NDV-Spotted Dove strain or its 100th-passaged

(Dove[100th]). Histopathological alterations were assessed at 3 dpi. Tissue samples were obtained from deceased chickens and analyzed using a high-powered microscope (HE ×20). Notable tissue lesions were identified in the NDV-Dove[100th] group, underscoring the increased virulence of the passaged strain (Fig. 2E). The pathological severity observed in the 100th passage on chicken was comparable to that of other avian viral infections. Infection with the NDV-Dove[100th] strain in chickens can result in multi-system damage. It can result in respiratory symptoms, including dyspnea and coughing, as well as gastrointestinal issues such as diarrhea and decreased appetite, with high mortality rates (Table 4). This strain, in severe cases, impacted the nervous system, leading to symptoms like ataxia and tremors, which indicate a high-severity pathology that significantly affects the health and survival of chickens. The virulence of the NDV-Dove[100th] strain appears to be at a medium-high level when compared to other common chicken-infecting viruses, such as Infectious Laryngotracheitis Virus (ILTV) and Chicken Anemia Virus (CAV) (35). ILTV primarily inflicts significant harm to the respiratory system, whereas CAV results in bone marrow hypoplasia and immunosuppression. The NDV-Dove[100th] strain exhibits a wider tissue tropism, impacting various organs and systems, indicating a significant pathogenic potential.

The passage of the Spotted Dove/China/08 virus was characterized through NGS to fully analyze its genome. The transcription, replication, virulence, and syncytia activity of the passage of Dove strain were notably modified, as previously reported (Fig. 1), resulting in enhanced virulence. Furthermore, through the application of NGS, multiple mutations were identified, and the impact of amino acid substitutions was analyzed within essential regions across various viral populations. These mutations affect the passage of the NDV-Dove strain through chicken embryos (Fig. 3A). To identify the key sites affecting viral passage and virulence by localizing differential amino acid positions, constructing six-point mutant strains, and rescuing them. We assessed membrane fusion activity, in vitro proliferation, pathogenicity, and performed experimental animal infections to identify mutations contributing to the enhanced virulence of the NDV-Dove strain. The results indicate that these mutations contribute to enhanced virulence, as shown by the ICPI data (Fig. 3; Table 3). The results showed that the virulence of the rDove[100th] strain in chickens correlates directly with the efficiency of viral genome replication (36). Notably, 25 amino acid positions on the P protein were identified as critical determinants of virulence in the NDV-Dove strain after 100 serial passages in chicken embryos. A recent study (14), on class I NDV strains in China, demonstrates that these viruses have gained virulence in poultry populations, showing an average

**TABLE 4** Mortality rates observed in the animal experiments (passage strain)[a]

| Factor | Impact on mortality rates | | Total |
|---|---|---|---|
| | Day 7 | Day 14 | |
| PBS | 0% | 0% | 0% |
| Dove strain | 0% | 0% | 0% |
| Dove[100th] | 70% | 20% | Near 100% |
| Infection route | Oculonasal | | |

[a]This table analyses the effects of various viruses on chicken mortality rates post-infection with distinct treatments at two time points (Day 7 and Day 14), utilizing the oculonasal infection route. The control group, treated with PBS, exhibited a mortality rate of 0% over the observed days. PBS is a neutral substance and is not pathogenic, establishing a baseline that indicates the experimental setup without a pathogenic agent does not result in mortality. The mortality rate in the Dove strain group is 0% at both Day 7 and Day 14. This suggests that the original form of the "Dove strain" of the virus, when administered via the oculonasal route, does not result in mortality in chickens during a 14-day observation period and exhibits low virulence for the test subjects under these conditions. At Day 7, the mortality rate in the group infected with Dove[100th] is 70%, while at Day 14, it decreases to 20%. The overall mortality rate approaches 100%. The data indicate that the passaged virus strain (Dove[100th]) exhibits high pathogenicity. The elevated mortality rate by Day 7 indicates a swift progression of the disease, while the further increase in mortality by Day 14 results in nearly complete mortality. This aligns with the heightened virulence demonstrated by the ICPI, MDT, and IVPI values in Table1, where the passaged strain exhibited improved virulence traits. The table indicates that the passaged Dove[100th] strain of the virus exhibits high pathogenicity and significant mortality, whereas the control (PBS) and the non-passaged Dove strain do not result in mortality under the specified experimental conditions involving oculonasal infection. This aligns with prior findings indicating heightened virulence in the passaged strain at the molecular and biological index levels (ICPI, MDT, and IVPI), demonstrating the in vivo mortality consequences of this increased virulence.

evolutionary rate of $1.797 \times 10^{-3}$ substitutions per site per year. Mutations were primarily found in the HN and P genes, suggesting that those genes may play a role in host range expansion and enhanced virulence. The construction of recombinant viruses with various P genes using reverse genetics technology reveals that the virulence of P gene-substituted strains is associated with the origin of the P gene. The findings indicate that the P gene is a contributing factor to the virulence of NDV, with its impact on virulence being influenced by multiple domains rather than a singular domain (37).

The P gene mutation (T25M) in paramyxoviruses may enhance virulence by modifying the structure, function, or interactions of the P protein, as identified in this study. The P protein serves multiple functions as a viral accessory protein, playing a critical role in viral replication, transcription, and immune evasion. Threonine (T) at position 25 is a polar amino acid characterized by a hydroxyl group, frequently serving as a phosphorylation site that modulates protein activity (14). In contrast, methionine (M) is a nonpolar, hydrophobic amino acid distinguished by its sulfur-containing side chain. This substitution may modify the local hydrophobicity and the secondary structure. The T25M mutation may eliminate a potential phosphorylation site during the passage, thereby disrupting regulatory signals that govern P protein function. The P protein functions as a cofactor for the viral RNA-dependent RNA polymerase (L protein) through its binding to the N protein. A structural modification at position 25 may enhance the stability of the P-N-L complex, thereby improving viral genome replication and transcription. The P protein of NDV inhibits IFN induction by disrupting the activation of transcription factors such as IRF3 and NF-κB (38). A mutated P protein (T25M) may enhance the suppression of IFN production, facilitating unchecked viral replication. The mutation may modify the P protein's capacity to engage with host immune receptors or signaling pathways, thereby diminishing the recruitment and activation of immune cells, such as macrophages and T cells, at the site of infection. *In vitro* mutagenesis experiments in NDV indicate that the T25M mutation in the P protein enhances viral replication in cell cultures, such as Vero cells (2) and chicken embryo fibroblasts, as demonstrated in the present study (Fig. 1A and 3B), and is associated with increased CPEs after 36 hpi in BHK-21 (Fig. 1D). *In vivo* studies using SPF chickens infected with T25M-mutated NDV strains may demonstrate increased mortality rates, reduced time to death, and more pronounced histopathological lesions, such as necrosis in the liver, spleen, and brain, in comparison to wild-type strains. Crystallographic analysis of the P protein, both with and without the T25M mutation, may elucidate alterations in its tertiary structure, including modifications in binding pockets or dimerization interfaces, that are associated with functional changes (39).

Evolution is driven by mutations within a population, which arise during the replication of nucleic acid molecules, and viral genomes undergo continuous mutation. The mutation spectra of viruses deviate from neutral distributions, containing components with unique biological properties that may otherwise go unnoticed. Viruses may undergo bottlenecks of differing intensities, contingent upon the number of genomes from a larger population that commence replication. A restricted number of genomes will lead to a limited variant repertoire for replication and adaptation to selective pressures. Measuring the intensity of intra-host bottlenecks would improve our comprehension of viral evolution within hosts. Bottlenecks add a layer of stochasticity that affects both short-term (within-host) and long-term (inter-host) evolutionary processes. Viruses that circulate in large populations within a specific environment typically undergo adaptation and enhancement suited to that environment (11, 40).

After the 100th serial passage, the NDV-Dove$^{100th}$ and the rDove$^{100th}$-P$_{T25M}$ mutant exhibited significantly enhanced virulence, as demonstrated by the ICPI tests in day-old chickens and the MDT in 9-day-old SPF chicken embryos (Tables 1 and 3). By contrast, the 369-position amino acid mutation on the P protein exhibited minimal impact on viral replication and virulence, with no notable alteration in quasispecies frequency. The 25-position amino acid mutation markedly influenced the virus's membrane fusion activity, replication, and overall virulence.

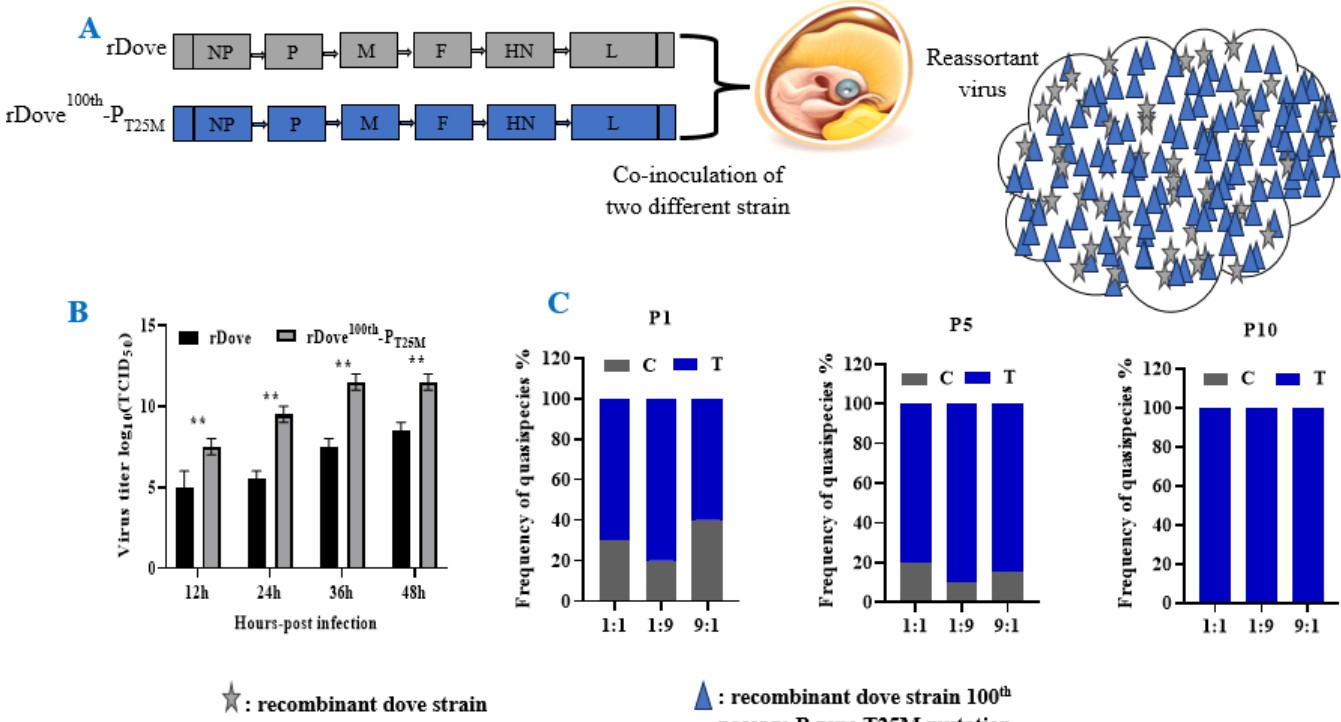

FIG 6 Generation and characterization of recombinant and reassortment virus. (A) The schematic illustrated the production of many viral recombinants by passage through chicken embryos. The recombinant Dove, characterized by a conventional viral genome NP, P, M, F, HN, and L proteins, together with rDove$^{100th}$-P$_{T25M}$ (a modified variant mutation T25M), is concurrently delivered into a host, SPF ECE. The current experiment relates to the co-inoculation of two distinct virus strains (rDove: rDove$^{100th}$-P$_{T25M}$) to examine their interactions when simultaneously introduced into a single environment (SPC ECE) for the purpose of assessing their effects. The co-inoculation of two distinct strains (rDove: rDove$^{100th}$-P$_{T25M}$) results in the creation of a reassortant virus. Reassortment viruses possessing segmented genomes (rDove: rDove$^{100th}$-P$_{T25M}$) exchanged genetic segments upon co-infection of a single cell by two distinct strains. This exchange results in a novel viral variation with a combination of genetic segments from the progenitor strains. The diagram of the reassortant virus illustrates the possible diversity produced by this process. (B) Chicken embryos TCID$_{50}$/mL in 9-day-old chicken embryos fixed at 48 hpi. The current findings indicate the time intervals post-infection, evaluated in hours (12 h, 24 h, 36 h, and 48 h), of ECE log10 (TCID$_{50}$). The logarithmic scale facilitates improved visibility of a broad spectrum of virus titer readings. At 12 hpi, the viral titer of rDove$^{100th}$-P$_{T25M}$ exceeds that of rDove. After 24 h, the disparity in titers between the two strains had risen, with rDove$^{100th}$-P$_{T25M}$ maintaining a superior titer. At 36 h and 48 h, rDove$^{100th}$-P$_{T25M}$ demonstrates a markedly elevated virus titer in comparison to rDove. The error bars indicate the variability in measurements at each time interval. The double asterisks (**) above the bars at each time point denote a statistically significant difference between the two strains, indicating that the observed variations in viral titers are unlikely to be attributable to chance. The persistent statistical significance (shown by **) at all time points underscores the dependability of the observed disparity in replication efficiency. The findings indicate that the genetic alteration in rDove$^{100th}$-P$_{T25M}$ significantly affects its replication characteristics relative to the rDove strain. $P < 0.01$. (C) The P protein is focused on NDV-Dove$^{100th}$-P$_{T25M}$, which has been cloned into pMD19-T to assess the frequency of fitness quasispecies. P1, P5, and P10 denote the first, fifth, and tenth passages, respectively. This study examined the impact of varying ratios of two viral components (1:1, 1:9, and 9:1) and two experimental conditions (C and T) on the frequency of viral quasispecies across multiple passages (P1, P5, and P10). The frequency of quasispecies is significant as it impacts the diversity of viral genes, subsequently influencing viral evolution, survival, and host adaptation. P1: Under a 1:1 ratio, the frequency of quasispecies in condition "C" exceeds that in condition "T." As the ratio transitions to 1:9 and 9:1, the disparity between "C" and "T" remains, with "C" typically exhibiting a greater frequency of quasispecies. P5: Consistent with P1, condition "C" exhibits a greater frequency of quasispecies than "T" across all ratios. The overall distribution has changed slightly compared to P1, suggesting an evolution or alteration in the quasispecies population over passages. P10: This passage indicates that for all ratios (1:1, 1:9, and 9:1), the frequency of quasispecies is almost the same between "C" and "T," implying that the disparity between the two conditions has decreased over multiple passages. The initial passages (P1 and P5) demonstrate a distinct variation in the frequency of quasispecies between the two conditions ("C" and "T") at various ratios. This indicates that the two experimental conditions influence the generation or maintenance of viral quasispecies populations at the experiment's outset. By P10, the differentiation between the two conditions has disappeared. The results indicate that over successive passages, additional factors may equalize the quasispecies frequencies, or that the initial disparities between "C" and "T" are progressively mitigated by the evolutionary dynamics of the viral population.

Fitness gains are the outcome of the competitive optimization of a viral quasispecies, characterized by the balance of mutation and selection within a specific environment

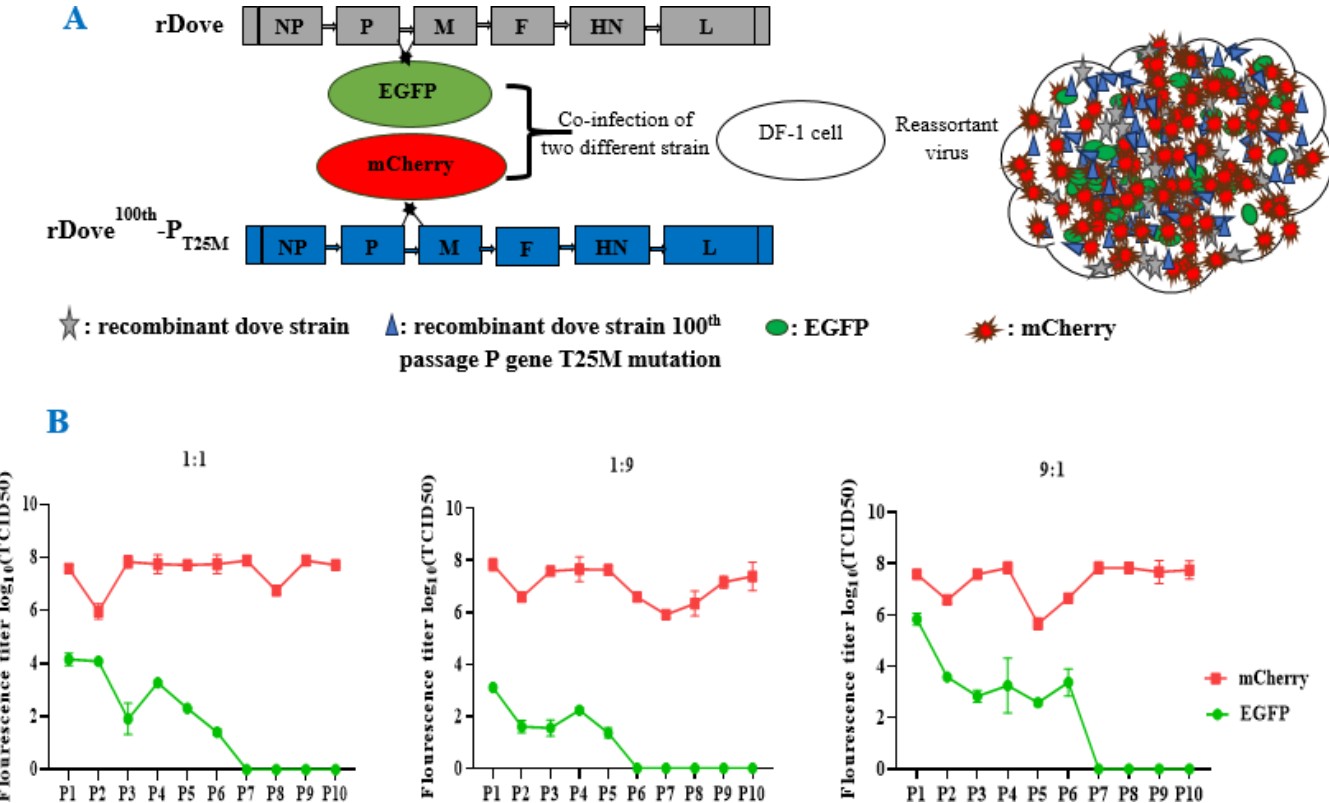

**FIG 7** Generation and characterization of rNDV-Dove strain expressing green and red fluorescence using in passage of virus in DF-1 cell. (A) The schematic represented the construction of over recombinants of the virus using in passage on DF-1 cell, and cells were infected with each virus at an MOI of 0.01 fixed at 48 hpi. The presence of different colored markers (green and red) in the resulting reassortant virus population indicates successful genetic recombination and the presence of both parental viral components in the new viruses. (B) Statistical analyses by different equal ratios (1:1, 1:9, and 9:1) of two viruses containing EGFP and mCherry fluorescence and the ratio of virus titers of each passage. We analyzed the balance between two virus strains (marked by EGFP and mCherry fluorescent proteins) that affect the brightness of their fluorescence over several rounds of growth (P1–P10). The dynamics of viral genetic components and their expression levels as the virus undergoes serial passages under varying strain ratios to provide insights into viral evolution, genetic stability, and the impact of initial strain ratios on long-term viral characteristics. From a 1:1 ratio, the mCherry-tagged virus starts with a high titer at P1, remains relatively stable with minor fluctuations until P6, then drops slightly at P7 and stabilizes again. The EGFP-tagged virus starts with a moderate titer, decreases at P2, increases at P3, and then shows a continuous decline from P4 onwards, reaching near-zero levels by P7–P10. In 1:9 ratios, the mCherry-tagged virus starts with a high titer at P1, shows some fluctuations but generally remains high until P6, then decreases slightly and stabilizes. The EGFP-tagged virus starts with a moderate titer, decreases at P2, has minor increases and decreases until P6, and then drops to near-zero levels by P7–P10. And on the ratio 9:1, the mCherry-tagged virus starts with a high titer at P1, decreases at P2, increases at P3, and then shows a general upward trend from P4 onwards. The EGFP-tagged virus starts with a moderate titer, decreases at P2, has minor fluctuations until P6, and then drops to near-zero levels by P7-P10. Across all ratios, the mCherry-tagged virus generally maintains a higher fluorescence titer over multiple passages compared to the EGFP-tagged virus. This suggests that the presence of the T25M mutant tagged viral component may have a replication in the cell. The different ratios do influence the titer trends, but the overall pattern of mCherry-tagged virus dominance remains consistent. For example, in the 9:1 ratio, the mCherry-tagged virus shows a more pronounced upward trend in later passages compared to the other ratios. The EGFP-tagged virus shows a consistent decline in titer over passages, indicating potential genetic instability or a competitive disadvantage against the mCherry-tagged virus. These findings are valuable for understanding the long-term behavior of genetically modified viruses and can inform research in areas such as viral vector design and the study of viral evolution under defined genetic conditions.

(41). The population size attained by a virus within an infected host can be substantial. While a single virus particle may not immediately initiate an infection or generate progeny, exposure to an altered environment can activate replication. Specific infectivity metrics, including plating efficiency, exhibit significant dependence on environmental factors (6). Following infection with a restricted quantity of particles, it is anticipated that a substantial population will exhibit increased fitness levels. Viruses that primarily replicate in a specific tissue are anticipated to improve their capacity to infect that tissue as the infection advances. Nonetheless, viruses that target multiple tissue types

may exhibit compartmentalization of fitness. Environmental changes or limitations on population size can hinder the process of fitness acquisition. The present study revealed a notable enhancement in virulence when comparing the parental NDV-Spotted Dove strain to its 100[th] passage derivative. The findings indicate that lentogenic strains with an artificial velogenic cleavage site do not attain the same virulence level as naturally derived velogenic strains.

Recombination generally takes place when two viruses concurrently infect a single host cell. In these instances, both viruses employ the cellular machinery to generate viral particles, leading to a combination of viral components, including newly synthesized genomes. Within the cell, several RNA viruses experience recombination or reassortment during their replication process (42, 43). In cells infected at a high multiplicity, the exchange of genetic material can enhance genetic diversity by combining previously distinct mutations into a single genome. The precise function of viral recombination in evolutionary processes is not fully understood; however, it is clear that these mechanisms can repair mutated genomes by eliminating harmful mutations, even in cells infected at low multiplicity. These "repairs" contribute to the preservation of viral fitness and may improve the mutational tolerance within viral populations. Recombination can occur in two primary ways under these conditions: first, homologous regions of viral genomes can pair and exchange segments, resulting in the physical breaking and rejoining of RNA or DNA. Second, viruses possessing segmented genomes can exchange complete segments through a mechanism referred to as reassortment (Fig. 6 and 7). Many studies have reported that in *vivo* and in *vitro*, avirulent viruses can acquire increased virulence, and viral quasispecies may gain virulence in the same environment when the virus is propagated in large populations. Interestingly, in the same environment (chicken embryos in this case), lentogenic viruses can evolve into virulent strains after 100 passages, characterized by specific mutation. We demonstrated that viral fitness within a quasispecies is influenced by the co-inoculation of a key mutation (rDove$^{100th}$-P$_{T25M}$) alongside the parental NDV Dove strain. Detection of Spotted Dove strain (rDove) in chicken embryos following 10 serial passages at different ratios (1:1, 1:9, and 9:1). Furthermore, the co-infection of recombinant strains labeled with green fluorescence and the Dove$^{100th}$-P$_{T25M}$ strain labeled with red fluorescence was observed in DF-1 cells. Specific primers targeting a 250-bp region, RNA was extracted and cloned into pMD19-T. The sequencing of the P gene (T25M) revealed complete genetic changes after 10 passages (Fig. 6C). Similar results were obtained when virus titers were assessed using a TCID$_{50}$/mL assay to compare the viral loads between the green and red fluorescence strains (Fig. 7B). The viral titer of the green-fluorescent strain was entirely reduced in comparison to the red-fluorescent strain. The absence of green fluorescence syncytia formation in the C group became apparent after 10 passages in chicken embryos and DF-1 cells. The F protein cleavage site of NDV has been identified as a key factor influencing virulence. In a persistent infection model using BHK-21 cells, it was observed that under conditions of competitive co-infection, the virulent NDV strain emerges as the dominant variant from the fifth passage. Conversely, in the absence of competition, both lentogenic and virulent NDV strains are sustained at comparable levels within the persistent infection cells. This indicates that environmental factors may significantly impact the fitness of NDV quasispecies (44). The dynamic changes of quasispecies confer significant genetic fitness to SARS-CoV-2 during intra-host evolution. The process can be accomplished by altering the genetic traits of essential functional genes, including the spike glycoprotein (45). The research indicates that the quasispecies structure of SARS-CoV-2 exhibits relative stability, characterized by a predominant mutant alongside several minor variants. Under conditions of high-pressure selection, minor mutants may gain a fitness advantage and subsequently become the predominant mutant. A study (46) investigating NDV quasispecies showed that modifications in the fusion cleavage site markedly affect the virulence of the virus. Nonetheless, the quasispecies diversity of NDV in natural infections and its role in virulence evolution remain largely unexplored.

## Conclusion

The strain exhibited increased virulence after 100 serial passages, demonstrating the evolutionary dynamics of RNA viruses. The primary factors contributing to this phenomenon are the large population size and the rapid evolution of the life cycle, which result in a faster pace than that of the host species. A weaker strain is known to be susceptible to it, according to principles of viral biology. Subsequent passages typically result in the ECE demonstrating increased virulence. During this investigation, six viruses exhibiting progressive mutations were identified in cell culture. Enhancement of viral replication in cell culture occurs without the establishment of a viral replicational profile in the absence of six progressive mutations. The six progressively modified viruses exhibited enhanced replication in cell culture compared to the control group.

## ACKNOWLEDGMENTS

This work was supported by the National Natural Science Foundation of China (No. 32373004).

## AUTHOR AFFILIATIONS

[1]College of Veterinary Medicine, Northwest A&F University, Yangling, Shaanxi, People's Republic of China
[2]Institut Supérieur du Développement Rural (ISDR), Université de Bangui BP 1450, Bangui, République Centrafricaine
[3]Département de Sciences de la vie, Faculté des Science (FS), Université de Bangui, Bangui, République Centrafricaine
[4]Shandong Animal Products Quality and Safety Center (Shandong Livestock and Poultry Slaughtering Technology Center), Shandong, People's Republic of China
[5]Qinghai University, Xining, People's Republic of China

## AUTHOR ORCIDs

Prince-Théodore Daguia-Wenam  http://orcid.org/0009-0009-2099-638X
Haijin Liu  http://orcid.org/0000-0002-6321-0844
Zengqi Yang  http://orcid.org/0000-0002-8230-490X

## FUNDING

| Funder | Grant(s) | Author(s) |
| --- | --- | --- |
| National Natural Science Foundation of China | No. 32373004 | Haijin Liu |

## AUTHOR CONTRIBUTIONS

Prince-Théodore Daguia-Wenam, Conceptualization, Data curation, Formal analysis, Methodology, Software, Visualization, Writing – original draft, Writing – review and editing | Kejia Lu, Data curation, Formal analysis, Validation, Writing – review and editing | Xueting Zhou, Data curation, Visualization, Writing – review and editing | Chuanqi Yan, Data curation, Writing – review and editing | Lina Tong, Data curation, Formal analysis, Writing – original draft, Writing – review and editing | Haijin Liu, Conceptualization, Methodology, Resources, Supervision, Writing – review and editing | Zengqi Yang, Conceptualization, Funding acquisition, Investigation, Methodology, Project administration, Resources, Supervision, Validation, Writing – review and editing

## DATA AVAILABILITY

The article contains all of the data that back up the study's findings.Full sequences generated were successfulsuccessfully submitted in GenBank with accession numbers: PX500221.1, PX500222.1, PX500223.1, and PX500224.1.

## ETHICS APPROVAL

The embryonated eggs were kept in our laboratory, while the chickens were housed at the Animal Laboratory Center, Northwest A&F University, Yangling, Shaanxi Province, People's Republic of China, with ethical reference number (IACUC2024-0605) with Biosafety Statement Level (ABSL-2). All experiments involving live viruses and animals were carried out in biosafety level (World Health Organization, https://www.who.int/publications/i/item/9789240011311).

## ADDITIONAL FILES

The following material is available online.

### Open Peer Review

**PEER REVIEW HISTORY (review-history.pdf).** An accounting of the reviewer comments and feedback.

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
