## [Reviewer comments · Microbiology Spectrum]

Microbiology Spectrum

The P Protein T25M substitution Is Involved in the Quasispecies and Virulence of Newcastle Disease Virus

Prince-Theodore Daguia-Wenam, Kejia Lu, Xueting Zhou, Chuanqi Yan, Lina Tong, Haijin Liu, and Zengqi Yang

Corresponding Author(s): Zengqi Yang, Northwest A&F University

Review Timeline:

Submission Date:	March 3, 2025
Editorial Decision:	May 23, 2025
Revision Received:	June 16, 2025
Editorial Decision:	August 27, 2025
Revision Received:	August 31, 2025
Editorial Decision:	October 7, 2025
Revision Received:	October 8, 2025
Editorial Decision:	October 23, 2025
Revision Received:	October 25, 2025
Accepted:	October 30, 2025

Editor: Rafael A. Medina

Reviewer(s): Disclosure of reviewer identity is with reference to reviewer comments included in decision letter(s). The following individuals involved in review of your submission have agreed to reveal their identity: Wentao Li (Reviewer #1); Ravindra P Veeranna (Reviewer #2)

Transaction Report:

DOI: <https://doi.org/10.1128/spectrum.00639-25>

Re: Spectrum00639-25 (**Newcastle Disease Virus Spotted Dove strain P gene T25M enhanced virulence in chicken after 100 serials passage through chicken embryos and its viral fitness quasispecies was determined by using NGS**)

Dear Prof. Zengqi Yang:

Thank you for the privilege of reviewing your work. You will see from the reviewers' comments that additional information needs to be provided. Specifically, please make sure to address the comment from reviewer #2 regarding the missing HI data, the suggestions on the animal experiments, provide information on cell line/s used, indicate the rationale for the points analyzed during passage selection, and discuss the potential mechanistic role of mutation T25M. Additionally, provide information on ethical approval and biosafety measures used during the performance of the experiments. Also, include the data and discuss all figures as indicated by reviewer #2. In your new version please revise the text for clarity, structure, and overall readability. We ask that this be provided, before we consider your manuscript further.

Revision Guidelines

Sincerely,
Rafael A. Medina
Editor
Microbiology Spectrum

Reviewer #1 (Comments for the Author):

The article titled "Newcastle Disease Virus Spotted Dove Strain P Gene T25M Enhances Virulence in Chickens After 100 Serial Passages Through Chicken Embryos, and Its Viral Fitness Quasispecies Was Determined Using NGS" by Prince-Théodore Daguia-Wenam et al. was submitted to Microbiology Spectrum. Using NGS sequencing technology, the authors explored the quasispecies phenomenon in the Newcastle Disease Virus (NDV) attenuated strain during serial passages in chicken embryos. They found that the NDV Dove strain's virulence was modified through viral quasispecies. By combining reverse genetics techniques, they identified that the change in virulence was primarily linked to the P protein at position 25, where the mutation resulted in an amino acid change from T to M. Before passage, the T amino acid variant at position 25 of the P protein accounted for 99.89%, while the M variant was only 0.05%. Over the course of the passages, the proportion of the M variant increased gradually, reaching 99.96% by the 100th passage, while the T variant dropped to 0.04%. Additionally, co-culture studies with chicken embryos and cells confirmed that the M variant at position 25 of the P protein had significantly stronger competitive ability than the T variant. These findings provide new insights into the viral quasispecies phenomenon in NDV and reveal a novel virulence determinant site, which has important implications for vaccine production and the development of attenuated vaccines.

While the article presents innovative findings and the results are robust enough to support the authors' conclusions, the writing does not meet the standard expected of scientific research papers and may be challenging for readers to follow. I recommend substantial revisions to improve clarity and readability. Specifically:

1. Some key results presented in the figures are not adequately discussed in the text (e.g., Fig. 3).
2. The logical flow of the manuscript could be improved. For example, the sections "Analysis of Viral Growth Kinetics of NDV-Dove Strain and Its Passage," "Pathogenicity Test Analysis," and "Effect of Different Passages of NDV-Dove Strain in Infected Chickens" all describe how the virus changes its cytopathic effects and pathogenicity after 100 passages in chickens. Combining these sections rather than presenting them separately would improve the flow of information.
3. Similar issues arise throughout the manuscript, particularly in the results section. I recommend that the authors carefully revise the manuscript to improve its clarity, structure, and overall readability.

Reviewer #2 (Comments for the Author):

Please refer to the attached document.

Summary

This study explores the evolution of virulence in a lentogenic Newcastle Disease Virus (NDV) strain originally isolated from a Spotted Dove, after 100 serial passages through chicken embryos. The authors identify the T25M mutation in the P gene as a key factor enhancing viral replication, pathogenicity, and overall viral fitness. Using recombinant NDV strains, in vivo pathogenicity indices (ICPI, MDT, IVPI), and next-generation sequencing (NGS), the study links quasispecies dynamics with increased virulence.

Major Comments

1. Reverse Genetic Confirmation of Mutation Effect
 - To establish causality, construct a recombinant NDV in which the T25M mutation in the P gene is reverted to the wild-type sequence. This will confirm whether the T25M mutation alone enhances replication, pathogenicity, and fitness.
2. Immune Response Analysis Missing
 - Although the manuscript mentions HI titers (lines 170–171), corresponding experimental data are not presented. Provide immunological data comparing antibody titers (e.g., HI assay), CD4/CD8 T-cell counts, or other immune markers between native and mutated NDV. These data are essential to support any conclusion regarding altered host immune response.
3. Mortality Data
 - Include mortality rates (if any) observed in the animal experiments. This would strengthen the virulence comparison across passage levels or recombinant strains.
4. Ethical Approval
 - Clearly state the ethical clearance reference number and the approving institutional animal ethics committee, especially for in vivo infection studies.
5. Biosafety Considerations
 - Provide details on biosafety containment measures employed to prevent accidental environmental release of the mutated NDV strains, especially given the increased virulence.
6. Mechanistic Explanation for P Gene's Role

- Discuss how the P gene mutation (T25M) mechanistically contributes to enhanced virulence, especially in light of established literature indicating that NDV virulence is primarily driven by the HN and F proteins (e.g., J. Virol. 2014; 88(15): 8696–8710). Clarify whether the P gene mutation has known or novel effects on viral polymerase activity, interferon antagonism, or host cell signaling.

7. Cell Line Comparison

- The manuscript reports increased replication and syncytia formation in BHK-21 cells. Were similar virulence assays (ICPI, MDT, IVPI) conducted in DF-1 cells to validate consistency across host systems?

8. Rationale for Passage Selection

- Justify the selection of passage points (e.g., P25, P50, P75, P100) for sequencing and phenotypic assays. Were these based on observed phenotypic shifts, genetic divergence, or predetermined intervals?

Minor Comments

1. Introduction Clarity

- Expand the introduction to briefly describe genes known to influence NDV virulence and host immune response. This will better align the background with the study's aim.

2. Cell Passage Information

- Mention the passage number of DF-1 and BHK-21 cells used in the assays to ensure cell line integrity and consistency.

3. Grammar and Style Corrections

- Line 392: “We were observed” → “We observed”
- Line 153: “Animal experimentally infection” → “Animal experiment”
- Line 34: “These reassortants viruses...” → “These reassortant viruses...”
- Ensure consistent notation of gene/protein names (e.g., “P gene,”)
- Please do corrections elsewhere in the manuscript.

4. Statistical Reporting

- P-values and statistical tests are not consistently reported. For example, Figures 2, 3, and 5 lack mention of statistical comparisons.

- For Figure 4, clarify how p-values were calculated for histopathological scoring. Was a scoring scale used? What statistical test was applied?

5. Figures

- Improve image resolution. Add arrows or labels to histological figures and viral plaques to guide interpretation.
- Enhance figure legends with more detail on sample sizes, treatment groups, and statistical significance.

6. Abbreviations

- Ensure all abbreviations are defined at first use (e.g., ICPI, IVPI, MDT, MOI, HI).

7. Statistical Methods

- Specify whether comparisons were made using t-tests, one-way ANOVA, or other statistical models for each dataset.

8. References

- Update and standardize references. Include guidelines from OIE/WHO for interpreting MDT, ICPI, and IVPI data. Some older citations may need replacement with more recent studies.

Reviewer #1 (Comments for the Author):

The article titled "Newcastle Disease Virus Spotted Dove Strain P Gene T25M Enhances Virulence in Chickens After 100 Serial Passages Through Chicken Embryos, and Its Viral Fitness Quasispecies Was Determined Using NGS" by Prince-Théodore Daguia-Wenam et al. was submitted to *Microbiology Spectrum*. Using NGS sequencing technology, the authors explored the quasispecies phenomenon in the Newcastle Disease Virus (NDV) attenuated strain during serial passages in chicken embryos. They found that the NDV Dove strain's virulence was modified through viral quasispecies. By combining reverse genetics techniques, they identified that the change in virulence was primarily linked to the P protein at position 25, where the mutation resulted in an amino acid change from T to M. Before passage, the T amino acid variant at position 25 of the P protein accounted for 99.89%, while the M variant was only 0.05%. Over the course of the passages, the proportion of the M variant increased gradually, reaching 99.96% by the 100th passage, while the T variant dropped to 0.04%. Additionally, co-culture studies with chicken embryos and cells confirmed that the M variant at position 25 of the P protein had significantly stronger competitive ability than the T variant. These findings provide new insights into the viral quasispecies phenomenon in NDV and reveal a novel virulence determinant site, which has important implications for vaccine production and the development of attenuated vaccines. While the article presents innovative findings and the results are robust enough to support the authors' conclusions, the writing does not meet the standard expected of scientific research papers and may be challenging for readers to follow. I recommend substantial revisions to improve clarity and readability. Specifically:

1. Some key results presented in the figures are not adequately discussed in the text (e.g., Fig. 3).

➤ It was corrected (**Fig3**), interpretation

2. The logical flow of the manuscript could be improved. For example, the sections "Analysis of Viral Growth Kinetics of NDV-Dove Strain and Its Passage," "Pathogenicity Test Analysis," and "Effect of Different Passages of NDV-Dove Strain in Infected Chickens" all describe how the

virus changes its cytopathic effects and pathogenicity after 100 passages in chickens. Combining these sections rather than presenting them separately would improve the flow of information.

- It was revised for clarity

3. Similar issues arise throughout the manuscript, particularly in the results section. I recommend that the authors carefully revise the manuscript to improve its clarity, structure, and overall readability.

- The manuscript, particularly in the results section was edited and rewritten

Summary

This study explores the evolution of virulence in a lentogenic Newcastle Disease Virus (NDV) strain originally isolated from a Spotted Dove, after 100 serial passages through chicken embryos. The authors identify the T25M mutation in the P gene as a key factor enhancing viral replication, pathogenicity, and overall viral fitness. Using recombinant NDV strains, in vivo pathogenicity indices (ICPI, MDT, IVPI), and next-generation sequencing (NGS), the study links quasispecies dynamics with increased virulence.

Major Comments

1. Reverse Genetic Confirmation of Mutation Effect

- To establish causality, construct a recombinant NDV in which the T25M mutation in the P gene is reverted to the wild-type sequence. This will confirm whether the T25M mutation alone enhances replication, pathogenicity, and fitness.

It was corrected “recombinant NDV T25M mutation was constructed” (line 255 to line 267) replication (**Fig4** “rT25M”), pathogenicity (**Table4**), and fitness (**Fig6 and 7**), were confirmed. In our manuscript the recombinant NDV with the T25M was named **rNDV-Dove^{100th}-P_{T25M}**.

2. Immune Response Analysis Missing

- Although the manuscript mentions HI titers (lines 170–171), corresponding experimental data are not presented. Provide immunological data comparing antibody titers (e.g., HI assay), CD4/CD8 T-cell counts, or other immune markers between native and mutated NDV. These data are essential to support any conclusion regarding altered host immune response.

It was completed to figure2 (**Fig.2.C**)

3. Mortality Data

- Include mortality rates (if any) observed in the animal experiments. This would strengthen the virulence comparison across passage levels or recombinant strains.

It was completed (**Table2 and 3**)

4. Ethical Approval

- Clearly state the ethical clearance reference number and the approving institutional animal ethics committee, especially for in vivo infection studies.

It was completed to line 145 to line 146

5. Biosafety Considerations

- Provide details on biosafety containment measures employed to prevent accidental environmental release of the mutated NDV strains, especially given the increased virulence.

It was corrected line 191 to line 195.

6. Mechanistic Explanation for P Gene's Role

- Discuss how the P gene mutation (T25M) mechanistically contributes to enhanced virulence, especially in light of established literature indicating that NDV virulence is primarily driven by the HN and F proteins (e.g., J. Virol. 2014; 88(15): 8696–8710). Clarify whether the P gene mutation has known or novel effects on viral polymerase activity, interferon antagonism, or host cell signaling.

It was corrected (Discussion line 912 to line 938)

7. Cell Line Comparison

- The manuscript reports increased replication and syncytia formation in BHK-21 cells. Were similar virulence assays (ICPI, MDT, IVPI) conducted in DF-1 cells to validate consistency across host systems?

It was completed (**Fig1. D. F, Fig3. A.B**)

8. Rationale for Passage Selection

- Justify the selection of passage points (e.g., P25, P50, P75, P100) for sequencing and phenotypic assays. Were these based on observed phenotypic shifts, genetic divergence, or predetermined intervals?

It was justified (line 205 to line 210)

Minor Comments

1. Introduction Clarity

- Expand the introduction to briefly describe genes known to influence NDV virulence and host immune response. This will better align the background with the study's aim.

It was corrected (line 59 to line 94)

2. Cell Passage Information

- Mention the passage number of DF-1 and BHK-21 cells used in the assays to ensure cell line integrity and consistency.

Passage number of DF-1 cell is mentioned in section “Co-infected DF-1 cell with different strain containing green and red protein” line 298 to line 311, (**Fig.7.B**); P1(first passage) to P10 (tenth passage). We haven't use BHK-21 to make the passage of the virus, but BHK-21 was used to determine the plaque forming units (PFU) line 157 to line 159 (**Fig1.E**)

3. Grammar and Style Corrections

- Line 392: “We were observed” → “We observed”

It was corrected to “We observed” line 494

- Line 153: “Animal experimentally infection” → “Animal experiment”

It was corrected to “Animal experiment”, line 182

- Line 34: “These reassortants viruses...” → “These reassortant viruses...”

It was edited and rewritten

- Ensure consistent notation of gene/protein names (e.g., “P gene,”)

It was corrected to “P gene”, line 44

- Please do corrections elsewhere in the manuscript.

The manuscript was edited and rewritten

4. Statistical Reporting

- P-values and statistical tests are not consistently reported. For example, Figures 2, 3, and 5 lack mention of statistical comparisons.

It was corrected (**Fig.1.C, F; 2.D; 3.B; 4.G-F and 6.B**), interpretation.

- For Figure 4, clarify how p-values were calculated for histopathological scoring. Was a scoring scale used? What statistical test was applied?

It was a mistake and corrected (**Fig.5**)

5. Figures

- Improve image resolution. Add arrows or labels to histological figures and viral plaques to guide interpretation.

It was corrected (**Fig.2.E and Fig.5**), interpretation

- Enhance figure legends with more detail on sample sizes, treatment groups, and statistical significance.

It was corrected (**Fig.6 and 7**)

6. Abbreviations

- Ensure all abbreviations are defined at first use (e.g., ICPI, IVPI, MDT, MOI, HI).

It was corrected (line 25 to line 27, line 202)

7. Statistical Methods

- Specify whether comparisons were made using t-tests, one-way ANOVA, or other statistical models for each dataset.

It was corrected. We used t-tests and one-way ANOVA (line 329)

8. References

- Update and standardize references. Include guidelines from OIE/WHO for interpreting MDT, ICPI, and IVPI data. Some older citations may need replacement with more recent studies.

It was updated(References)

Re: Spectrum00639-25R1 (**Newcastle Disease Virus Spotted Dove strain P gene T25M enhanced virulence in chicken after 100 serials passage through chicken embryos and its viral fitness quasispecies was determined by using NGS**)

Dear Prof. Zengqi Yang:

Thank you for the privilege of reviewing your work. Below you will find my comments, instructions from the Spectrum editorial office, and the reviewer comments.

You will see from the referees' comments that additional information needs to be provided. Please revise the manuscript to address the comments of reviewer #2. Specifically, please provide mortality data and HA/HI titers comparisons as suggested by the reviewer.

Revision Guidelines

Sincerely,
Rafael A. Medina
Editor
Microbiology Spectrum

Reviewer #2 (Comments for the Author):

The authors have addressed my previous comments satisfactorily and provided additional experimental data. However, I would

still like to see mortality data and HA/HI titre comparisons between rNDV, the parental strain, and NDV-Dove100th (rDove100th-PT25M). Such comparative analysis will be crucial to confirm whether the P gene mutation contributes directly to the observed increase in virulence, and will provide a stronger experimental foundation for the author's conclusions.

Reviewer#2 (Comments for the Author):

The authors have addressed my previous comments satisfactorily and provided additional experimental data. However, I would still like to see mortality data and HA/HI titre comparisons between rNDV, the parental strain, and NDV-Dove100th (rDove100th-PT25M). Such comparative analysis will be crucial to confirm whether the P gene mutation contributes directly to the observed increase in virulence, and will provide a stronger experimental foundation for the author's conclusions.

✚ Mortality data: It was revised, (Table2 and 3).

Table 2: Mortality rates observed in the animal experiments (passage strain)

Factor	Impact on mortality rates		Total
	Day 7	Day 14	
PBS	0%	0%	0%
Dove strain	0%	0%	0%
Dove^{100th}	70%	20%	Near 100%
Infection route	Oculonasal		

Interpretation:

This table analyzes the effects of various viruses on chicken mortality rates post-infection with distinct treatments at two time points (Day 7 and Day 14), utilizing the oculonasal infection route. The control group, treated with phosphate buffered saline (PBS), exhibited a mortality rate of 0% over the observed days. PBS is a neutral substance and is not pathogenic, establishing a baseline that indicates the experimental setup without a pathogenic agent does not result in mortality. The mortality rate in the Dove strain group is 0% at both Day 7 and Day 14. This suggests that the original form of the “Dove strain” of the virus, when administered via the oculonasal route, does not result in mortality in chickens during a 14-day observation period and exhibits low virulence for the test subjects under these conditions. At Day 7, the mortality rate in the group infected with Dove^{100th} is 70%, while at Day 14, it decreases to 20%. The overall mortality rate approaches 100%. The data indicate that the passaged virus strain (Dove^{100th}) exhibits high pathogenicity. The elevated mortality rate by Day 7 indicates a swift progression of the disease,

while the further increase in mortality by Day 14 results in nearly complete mortality. This aligns with the heightened virulence demonstrated by the ICPI, MDT, and IVPI values in the previously table1, where the passaged strain exhibited improved virulence traits. The table indicates that the passaged Dove^{100th} strain of the virus exhibits high pathogenicity and significant mortality, whereas the control (PBS) and the non-passaged Dove strain do not result in mortality under the specified experimental conditions involving oculonasal infection. This aligns with prior findings indicating heightened virulence in the passaged strain at the molecular and biological index levels (ICPI, MDT, and IVPI), demonstrating the in vivo mortality consequences of this increased virulence.

Table 3: Mortality rates observed in the animal experiments (recombinant strain)

Factor	Impact on mortality rates		Total
	Day 7	Day 14	
PBS	0%	0%	0%
rDove strain	0%	0%	0%
rDove ^{100th} -P _{T25M}	40%	20%	60%
rDove ^{100th} -P _{S369N}	30%	0%	30%
rDove ^{100th} -M _{S200N}	0%	0%	0%
rDove ^{100th} -HN _{N147D}	0%	0%	0%
rDove ^{100th} -HN _{I405L}	0%	20%	20%
rDove ^{100th} -HN _{K495N}	10%	0%	10%
Infection route	Oculonasal		

Interpretation:

This table demonstrates data about the effects of various viral strains (mutants of a passaged Dove) on the mortality rates of test subjects (chickens) when infected through the oculonasal route. Phosphate buffered saline (PBS) used as a negative control: The absence of mortality at

Day 7, Day 14, and overall indicates that the buffer is non-pathogenic and does not induce fatalities in the test system. The rDove strain also functions as a control. A 0% mortality rate shows that this strain does not induce mortality in chickens within the 14-day observation period, and the virus is classified as avirulent. rDove^{100th}-P_{T25M} has a 40% mortality rate at Day 7 and 20% at Day 14, resulting in an overall mortality rate of 60%. This suggests that the mutant strain is pathogenic. The high mortality (Day 7) indicates a swift pathogenic mechanism, and the persistent mortality by Day 14 results in a considerable overall mortality rate. The T25M mutation may correlate with heightened pathogenicity. rDove^{100th}-P_{S369N} exhibits a 30% mortality rate at Day 7 and 0% at Day 14, resulting in an overall mortality rate of 30%. Mortality is predominantly observed at Day 7. The mutation at position S369N may enhance virulence; nevertheless, the host response could potentially regulate the infection by Day 14, leading to no further death. rDove^{100th}-M_{S200N} exhibits 0% mortality at Day 7, Day 14, and overall. This indicates that the mutation at position S200M in this recombinant, passaged viral strain did not produce a pathogenic phenotype in the experimental setting. The mutation may not influence critical virulence factors or could potentially be detrimental to the virus. rDove^{100th}-HN_{N147D} with a mortality rate of 0% at all time periods, the mutation at position 147 in the HN did not result in enhanced virulence. rDove^{100th}-HN_{I405L} exhibits 0% mortality at Day 7, escalating to 20% by Day 14, resulting in an overall mortality rate of 20%. The delayed mortality seen alone on Day 14 indicates that the mutation at position 405 in the HN protein may lead to a more gradual pathogenic mechanism. The virus may multiply or inflict damage progressively, resulting in fatality at a subsequent time point. rDove^{100th}-HN_{K495N} exhibits a mortality rate of 10% at Day 7 and 0% at Day 14, resulting in an overall mortality rate of 10%. The mutation at position 495 in the HN protein results in reduced virulence, allowing the host to potentially mitigate further death by Day 14.

HA titers: It was completed. (Fig.2.A and Fig.4.H)

HA titers: It was completed. (Fig.4.I)

Re: Spectrum00639-25R2 (**Newcastle Disease Virus Spotted Dove strain P gene T25M enhanced virulence in chicken after 100 serials passage through chicken embryos and its viral fitness quasispecies was determined by using NGS**)

Dear Prof. Zengqi Yang:

Thank you for the privilege of reviewing your work. Below you will find my comments, instructions from the Spectrum editorial office, and the reviewer comments.

I am pleased to inform you that your manuscript has been editorially accepted for publication. However, the current title is quite long and could be improved by focusing more on the key findings rather than the techniques used in the study. I would appreciate it if you could submit a revised version of the title. Once these are completed, please return your submission so that I can move your paper forward to acceptance.

Revision Guidelines

Sincerely,
Rafael A. Medina
Editor
Microbiology Spectrum

Reviewer #2 (Comments for the Author):

The authors have addressed all my comments. The revision has strengthened the scientific evidence that rDove100th-PT25M directly contributes to the increased virulence of the NDV Spotted Dove strain.

Re: Spectrum00639-25R3 (The 25th Amino Acid of the P Protein Is Involved in the Quasispecies and Virulence of Newcastle Disease Virus)

Dear Prof. Zengqi Yang:

Thank you for the privilege of reviewing your work. Below you will find my comments and instructions from the Spectrum editorial office.

I am pleased to inform you that your manuscript has been editorially accepted for publication. However, I would like to suggest a slightly modified title: "The P Protein T25M substitution Is Involved in the Quasispecies and Virulence of Newcastle Disease Virus". Once you approve I will move your paper forward to acceptance.

Revision Guidelines

Sincerely,
Rafael A. Medina
Editor
Microbiology Spectrum

Reviewer#2 (Comments for the Author):

The authors have addressed my previous comments satisfactorily and provided additional experimental data. However, I would still like to see mortality data and HA/HI titre comparisons between rNDV, the parental strain, and NDV-Dove100th (rDove100th-PT25M). Such comparative analysis will be crucial to confirm whether the P gene mutation contributes directly to the observed increase in virulence, and will provide a stronger experimental foundation for the author's conclusions.

✚ Mortality data: It was revised, (Table2 and 3).

Table 2: Mortality rates observed in the animal experiments (passage strain)

Factor	Impact on mortality rates		Total
	Day 7	Day 14	
PBS	0%	0%	0%
Dove strain	0%	0%	0%
Dove^{100th}	70%	20%	Near 100%
Infection route	Oculonasal		

Interpretation:

This table analyzes the effects of various viruses on chicken mortality rates post-infection with distinct treatments at two time points (Day 7 and Day 14), utilizing the oculonasal infection route. The control group, treated with phosphate buffered saline (PBS), exhibited a mortality rate of 0% over the observed days. PBS is a neutral substance and is not pathogenic, establishing a baseline that indicates the experimental setup without a pathogenic agent does not result in mortality. The mortality rate in the Dove strain group is 0% at both Day 7 and Day 14. This suggests that the original form of the “Dove strain” of the virus, when administered via the oculonasal route, does not result in mortality in chickens during a 14-day observation period and exhibits low virulence for the test subjects under these conditions. At Day 7, the mortality rate in the group infected with Dove^{100th} is 70%, while at Day 14, it decreases to 20%. The overall mortality rate approaches 100%. The data indicate that the passaged virus strain (Dove^{100th}) exhibits high pathogenicity. The elevated mortality rate by Day 7 indicates a swift progression of the disease,

while the further increase in mortality by Day 14 results in nearly complete mortality. This aligns with the heightened virulence demonstrated by the ICPI, MDT, and IVPI values in the previously table1, where the passaged strain exhibited improved virulence traits. The table indicates that the passaged Dove^{100th} strain of the virus exhibits high pathogenicity and significant mortality, whereas the control (PBS) and the non-passaged Dove strain do not result in mortality under the specified experimental conditions involving oculonasal infection. This aligns with prior findings indicating heightened virulence in the passaged strain at the molecular and biological index levels (ICPI, MDT, and IVPI), demonstrating the in vivo mortality consequences of this increased virulence.

Table 3: Mortality rates observed in the animal experiments (recombinant strain)

Factor	Impact on mortality rates		Total
	Day 7	Day 14	
PBS	0%	0%	0%
rDove strain	0%	0%	0%
rDove^{100th}-P_{T25M}	40%	20%	60%
rDove^{100th}-P_{S369N}	30%	0%	30%
rDove^{100th}-M_{S200N}	0%	0%	0%
rDove^{100th}-HN_{N147D}	0%	0%	0%
rDove^{100th}-HN_{I405L}	0%	20%	20%
rDove^{100th}-HN_{K495N}	10%	0%	10%
Infection route	Oculonasal		

Interpretation:

This table demonstrates data about the effects of various viral strains (mutants of a passaged Dove) on the mortality rates of test subjects (chickens) when infected through the oculonasal route. Phosphate buffered saline (PBS) used as a negative control: The absence of mortality at

Day 7, Day 14, and overall indicates that the buffer is non-pathogenic and does not induce fatalities in the test system. The rDove strain also functions as a control. A 0% mortality rate shows that this strain does not induce mortality in chickens within the 14-day observation period, and the virus is classified as avirulent. rDove^{100th}-P_{T25M} has a 40% mortality rate at Day 7 and 20% at Day 14, resulting in an overall mortality rate of 60%. This suggests that the mutant strain is pathogenic. The high mortality (Day 7) indicates a swift pathogenic mechanism, and the persistent mortality by Day 14 results in a considerable overall mortality rate. The T25M mutation may correlate with heightened pathogenicity. rDove^{100th}-P_{S369N} exhibits a 30% mortality rate at Day 7 and 0% at Day 14, resulting in an overall mortality rate of 30%. Mortality is predominantly observed at Day 7. The mutation at position S369N may enhance virulence; nevertheless, the host response could potentially regulate the infection by Day 14, leading to no further death. rDove^{100th}-M_{S200N} exhibits 0% mortality at Day 7, Day 14, and overall. This indicates that the mutation at position S200M in this recombinant, passaged viral strain did not produce a pathogenic phenotype in the experimental setting. The mutation may not influence critical virulence factors or could potentially be detrimental to the virus. rDove^{100th}-HN_{N147D} with a mortality rate of 0% at all time periods, the mutation at position 147 in the HN did not result in enhanced virulence. rDove^{100th}-HN_{I405L} exhibits 0% mortality at Day 7, escalating to 20% by Day 14, resulting in an overall mortality rate of 20%. The delayed mortality seen alone on Day 14 indicates that the mutation at position 405 in the HN protein may lead to a more gradual pathogenic mechanism. The virus may multiply or inflict damage progressively, resulting in fatality at a subsequent time point. rDove^{100th}-HN_{K495N} exhibits a mortality rate of 10% at Day 7 and 0% at Day 14, resulting in an overall mortality rate of 10%. The mutation at position 495 in the HN protein results in reduced virulence, allowing the host to potentially mitigate further death by Day 14.

HA titers: It was completed. (Fig.2.A and Fig.4.H)

HA titers: It was completed. (Fig.4.I)

Re: Spectrum00639-25R4 (The P Protein T25M substitution Is Involved in the Quasispecies and Virulence of Newcastle Disease Virus)

Dear Prof. Zengqi Yang:

I am pleased to inform you that your manuscript has been accepted, and I am forwarding it to the ASM production staff for publication. Your paper will first be checked to make sure all elements meet the technical requirements. ASM staff will contact you if anything needs to be revised before copyediting and production can begin. Otherwise, you will be notified when your proofs are ready to be viewed.

Sincerely,
Rafael A. Medina
Editor
Microbiology Spectrum